# LABEL INFORMATIVENESS-BASED MINORITY OVER-SAMPLING IN GRAPHS (LIMO)

## ABSTRACT

Class imbalance is a pervasive issue in many real-world datasets, particularly in graph-structured data, where certain classes are significantly underrepresented. This imbalance can severely impact the performance of Graph Neural Networks (GNNs), leading to biased learning or over-fitting. The existing oversampling techniques often overlook the intrinsic properties of graphs, such as Label Informativeness (LI), which measures the amount of information a neighbor's label provides about a node's label. To address this, we propose Label Informativeness-based Minority Oversampling (LIMO), a novel algorithm that strategically oversamples minority class nodes by augmenting edges to maximize LI. This technique generates a balanced, synthetic graph that enhances GNN performance without significantly increasing data volume. Our theoretical analysis shows that the effectiveness of GNNs is directly proportional to label informativeness, with mutual information as a mediator. Additionally, we provide insights into how variations in the number of inter-class edges influence the LI by analyzing its derivative. Experimental results on various homophilous and heterophilous benchmark datasets demonstrate the effectiveness of LIMO in improving the performance of node classification for different imbalance ratios, with particularly significant improvements observed in heterophilous graph datasets. Our code is available at `https://anonymous.4open.science/r/limo-12CC/`

## 1 INTRODUCTION

The emergence of Graph Neural Networks (GNNs) has pushed the boundaries of graph structure analysis (Joshi & Mishra, 2021). These networks harness node attributes and graph topology to enhance learning outcomes. Approaches like Graph Convolutional Networks (GCNs) and Graph Attention Networks (GATs) have shown marked improvements in tasks such as node classification and link prediction (Kipf & Welling, 2017; Velickovic et al., 2018). By utilizing both node features and edge information, these methodologies capture intricate relationships within graphs, thereby boosting performance on graph tasks (Hamilton et al., 2017).

Class imbalance is a prevalent issue in many real-world datasets Kim et al. (2020), where certain classes are significantly underrepresented compared to others in a dataset. That is, the number of samples belonging to one class (the majority class) far exceeds that of another (the minority class). Consider an example of classifying medical images for a certain disease. The classifier tends to fail to precisely classify if the dataset is skewed towards any one of the positive or negative classes for the patient having the disease Tasci et al. (2022). The classifier is likely to be biased towards predicting the class labels of the majority of the images in the dataset. Such imbalance skews the performance of machine learning models, and the model tends to favor the majority class because it dominates the training process He & Garcia (2009). This is particularly problematic when the minority class represents rare but critical cases, such as fraud or disease detection Batista et al. (2004). It becomes difficult to use traditional machine learning algorithms while working with a class-imbalanced dataset because they often assume an equal distribution of classes. In the presence of class imbalance, the algorithms are likely to give biased predictions Shwartz-Ziv et al. (2024). Specifically, models may achieve high overall accuracy by simply predicting the majority class more frequently, but their performance on the minority class remains poor.

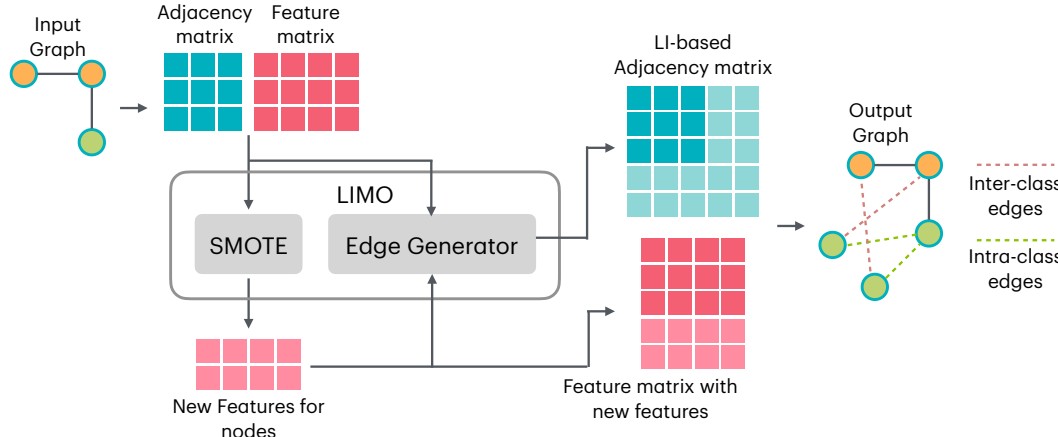

Figure 1: LIMO: The procedure initiates with an input graph, represented by its adjacency and feature matrices. Synthetic Minority Oversampling Technique (SMOTE) Chawla et al. (2002) and Edge Generator are employed to interpolate new features for minority class nodes and strategically add edges, maximizing Label Informativeness (LI) respectively. This process involves both inter-class and intra-class edge additions based on LI optimization criteria. The resulting balanced graph enhances minority class representation, leading to improved Graph Neural Network (GNN) performance in classification tasks.

Imbalanced node classification presents significant challenges for existing Graph Neural Networks (GNNs). In scenarios where the majority class dominates, the loss function becomes skewed, causing the GNN to overfit to the majority class while neglecting the minority class. This leads to poor predictive performance on minority class samples, limiting the effectiveness of GNNs in real-world applications characterized by imbalanced class distributions, such as malicious account detection. Addressing this issue is crucial for improving the adoption of GNNs in such tasks. Many previous works have tried addressing these challenges Chen et al., 2021; Zhao et al., 2021; Ashmore & Chen, 2023; Wang et al., 2022b; Hsu et al., 2024. These approaches either add synthetic nodes based on the features of the existing nodes in the graph or train the node classifier to learn with the class-imbalanced dataset.

In our work, we take a different approach to overcome the class imbalance. Our algorithm LIMO uses the concept of label informativeness of the given graph to mitigate the issue of class imbalance by strategically adding the edges to the graph. Empirically, it has been established that a positive correlation exists between LI and model performance Platonov et al. (2024). In our work, we further extend it and formally establish the positive correlation. In general, increasing the LI of the graph increases model performance. Hence, we add the synthetically generated nodes and edges to improve the graph's label informativeness. Additionally, LIMO acts on the class imbalanced dataset before it is given as an input to the GNN, thereby reducing the overhead cost of training the classifier to learn on the imbalanced dataset.

Our main **contributions** are: First, we propose Label Informativeness-based Minority Oversampling (LIMO). We theoretically establish the relationship between the label informativeness and the accuracy of the model predictions in GNN. Additionally, we analyze the influence of change in the number of inter-class and intra-class edges on LI. Finally, we empirically validate our proposed method on the node classification task and observe that it outperforms state-of-the-art approaches by a significant margin. Figure 1 gives us an illustration of how different components of LIMO help in generating the synthetic node features and edges the graph to include the synthetic nodes.

## 2 BACKGROUND

### 2.1 CLASS-IMBALANCE IN GRAPHS

We represent a graph as $G = \{\mathcal{V}, E, F, Y\}$, where $\mathcal{V} = \{v_1, \ldots, v_n\}$ comprises a set of $n$ nodes. $E$ is the set of edges in the graph. The adjacency matrix corresponding to the graph $G$ is denoted by $A \in \mathbb{R}^{n \times n}$, while $F \in \mathbb{R}^{n \times d}$ signifies the node feature matrix. $f_v \in \mathbb{R}^{1 \times d}$ represents the $d$ dimensional features of node $v$. The class information for nodes in $G$ is represented by $Y \in \mathbb{R}^n$. The class label of a node $v$ is represented as $y_v$. During the training phase, only a portion of $Y$, labeled as $Y_L$, is accessible, containing labels for a subset of nodes, $V_L$. The total number of classes is $C$, denoted as $\{0, 1, \ldots, C-1\}$. The Imbalance Ratio (IR) in the graph context can be expressed as:

$$\text{IR} = \frac{\min_y n_y}{\max_y n_y} \tag{1}$$

where $\min_y n_y$ and $\max_y n_y$ represent the number of nodes in the minority and majority classes, respectively, where $y \in \{1, 2, \ldots C\}$. A low IR indicates a significant imbalance, resulting in biased models that favor the majority class and underperform on the minority class.

### 2.2 LABEL INFORMATIVENESS

Label informativeness (LI) in a graph Platonov et al. (2024) measures how much information the label of a node provides about the label of its neighbor. According to Platonov et al. (2024), Label Informativeness (LI) serves as a complementary measure to homophily, emphasizing the predictive power of neighboring labels. This shows a strong correlation with Graph Neural Network (GNN) performance, even in heterophilous graph structures. It can be defined using mutual information $I(Y_u; Y_v)$ between the labels $y_u$ and $y_v$ of connected nodes $u$ and $v$:

$$\text{LI}(G) = 2 - \frac{\sum_{c_1, c_2} p(c_1, c_2) \log p(c_1, c_2)}{\sum_c \bar{p}(c) \log \bar{p}(c)} \tag{2}$$

where

$$p(c_1, c_2) = \frac{\sum_{(u,v) \in E} \mathbf{1}\{y_u = c_1, y_v = c_2\}}{2|E|} \tag{3}$$

and

$$\bar{p}(c) = \frac{D_c}{2|E|} \tag{4}$$

where $c_1$ and $c_2$ denote the labels of the nodes in the graph. Specifically $c_1$ represents the label of node $u$ and $c_2$ represents the label of node $v$. Both $u$ and $v \in E$ where $E$ is the edges set of the graph. Additionally, $D_c$ refers to the total degree of all the nodes present in class $c$. The LI of a graph, denoted as $\text{LI}(G)$, increases when edges within the same class are added, as this enhances the predictive capability of neighboring nodes for label determination. Conversely, the addition of edges between different classes reduces LI by diminishing the predictive strength of neighboring labels.

## 3 PROBLEM STATEMENT

Consider a graph $G$, that exhibits a significant class imbalance, i.e. the IR as per equation 1 is significantly low. This imbalance can lead to biased learning or overfitting in GNNs, resulting in poor performance, especially for underrepresented classes. Our goal is to synthetically add the nodes, edges, features, and labels to the imbalanced graph such that the label informativeness of the newly formed graph increases. More formally, we first generate the synthetic features using Synthetic minority oversampling technique (SMOTE) Chawla et al. (2002) for the minority class, and then we aim to find the following:

$$\underset{E'}{\arg\max} \ \text{LI}(G)$$

where $E'$ is the set of newly generated edges. In this way, we reduce the class imbalance by exploring and leveraging the relationship between the performance of the GNNs and the label informativeness. Thereby improving GNN performance for the task of node classification. We verify the enhanced performance on homophilous and heterophillous graph datasets.

# 4 LABEL INFORMATIVENESS BASED MINORITY OVERSAMPLING (LIMO)

The LIMO algorithm addresses class imbalance in graph data by generating synthetic nodes for minority classes, improving the representation of underrepresented classes to enhance machine learning model performance. A graph $G$ is given as input to the algorithm, and it outputs a modified, balanced graph $G' = \{\mathcal{V},' A,' F,' Y'\}$. To achieve this, the minority classes $c_i$s are first identified within the dataset. A matrix $P$ is then defined to quantify the distribution of edges over the classes, where $P = [p(c_1, c_2)]_{c_1,c_2=0}^{C-1}$ and $p(c_1, c_2)$ is computed using equation 3. For each node $v$ that belongs to a minority class, and for its nearest neighbor $u$ in the same class, i.e. $y_v$, and a synthetic node $s$ with feature vector $f_s$ is created using SMOTE, with $f_s = f_v + \lambda(f_u - f_v)$, where $\lambda$ is a random value between 0 and 1. We call this set of newly generated node features $S$. This set contains the nodes that belong to the class $y_v$.

These nodes are added to the vertex set, and it is updated as $\mathcal{V}' = \mathcal{V} \cup S$. Subsequently, the feature set and the set of class labels are updated as $F' = F \cup \{f_s\}$, and $Y' = Y \cup \{y_s\}$ respectively, for $s \in S$, and where $f_s$ is the feature vector generated by SMOTE. We update the adjacency matrix to include the synthetic nodes $S$ based on the condition that increases the LI of the graph, which in turn improves the performance of the GNN model. We add the inter-class edges between all synthetic nodes of the minority class and the rest of the classes, as well as intra-class edges among the nodes of the minority class, based on the criteria provided in theorem 1.

Homophily is not truly necessary for good GNN performance. Certain types of "good" heterophily exist, under which GCNs can achieve strong performance Ma et al. (2023). According to Platonov et al. (2024) the Spearman correlation coefficient between accuracy and LI is more than the Spearman correlation coefficient between accuracy and homophily. This was the motivation behind using LI to mitigate the class imbalance problem.

**Theorem 1.** *Let $e_{bc}$ be the number of inter-class edges for class $c_1$ and $c_2$ and $e_{wc}$ be the number of intra-class edges for class $c_1$ in a graph, where $c_1$ and $c_2$ are classes in the graph. If classes $c_1$ and $c_2$ satisfy the condition $e_{bc} \cdot 1.31167627 > e_{wc}$, then adding all inter-class edges between $c_1$ and $c_2$ to the graph i.e. increasing $e_{bc}$, and adding intra-class edges in $c_1$ i.e., increasing $e_{wc} < 1.31167627 \cdot e_{bc}$ will increase the Label Informativeness (LI).*

The proof is described in Appendix A.1

Specifically, at the node level, an edge $(s, w)$ is added between the new node $s$ and the node $w \in \mathcal{V}$ if the two conditions as mentioned in theorem 1 are satisfied i.e. $y_s \neq y_w$ (inter-class) and $P(y_s, y_w) > \frac{1}{t} \times P(y_s, y_s)$ or $y_s = y_w$(intra-class) and $P(y_s, y_s) > t \times \sum_{i=0}^{C-1} P(y_s, y_i)$, where $w$ is an existing node in $\mathcal{V}$. Here, $t$ takes the value 1.31167627 according to theorem 1. The above condition is obtained by dividing the condition mentioned in theorem 1 by a constant $(e_{bc} + e_{wc})$ and substituting $\frac{e_{wc}}{e_{bc}+e_{wc}} = P(y_s, y_s)$ and $\frac{e_{bc}}{e_{bc}+e_{wc}} = P(y_s, y_w)$ where $e_{bc}$ and $e_{wc}$ are the number of inter-class edges and intra-class edges respectively in the graph. If either condition is satisfied, the edge $(s, w)$ is added, updating the adjacency matrix as $A' = A \cup \{(s, w)\}$. This method, described in algorithm 1, balances class distributions in graph data while maintaining the graph's structure and increasing its LI, leading to better performance for node classification using GNN.

Impact of $t$ on model performance:

- For any other threshold $t' < t$, adding intra-class edges such that their count falls within $[t', t]$ times the total intra-class edges for a specific class will decrease the LI.

- Similarly, if $t' > t$, adding inter-class edges between the minority class and another class, with their count falling within $[t, t']$ times the total intra-class edges of the minority class, will also decrease the LI.

As LIMO aims at increasing the LI to improve model performance, we add the edges as per theorem 1.

---

**Algorithm 1**

---

1: **Input:** $G = \{\mathcal{V}, A, F, Y\}$
2: **Output:** return $G' = \{\mathcal{V}', A', F', Y'\}$ (Balanced)
3: Identify the minority classes and the nodes in those classes
4: $P \leftarrow [p(c_1, c_2)]_{c_1, c_2 = 0}^{C-1}$, where $c_1$ and $c_2$ are classes and $p(c_1, c_2)$ is calculated using eq(3)
5: **for** All minority nodes v **do**
6:    Find the nearest neighbor u $i \in y_v$ using eq(6)
7:    Interpolate between u and v using eq(7) to create a synthetic node s
8:    Add s to the vertex set to get $\mathcal{V}'$
9:    Add feature of s to the features set to get $F'$
10:    Assign the $y_s \leftarrow y_v$ and implement it in $Y'$
11:    **for** All nodes, w $\in \mathcal{V}$-{v} **do**
12:       **if** ($y_s \neq y_w$ and P($y_s$, $y_w$) > (1/t)$\times$ P($y_s, y_s$)) or
       (($y_s = y_w$ and P($y_s, y_s$) > t$\times \sum_{i=0}^{C-1}$ P($y_s, y_i$)) **then**
13:          Add edge (s,w) to adjacency matrix to get $A'$
14:       **end if**
15:    **end for**
16: **end for**
17: **return** $G' = \{\mathcal{V}', A', F', Y'\}$

---

### 4.1 Interpretation of Label Informativeness (LI) Differentiation

In our study, we consider the LI of a subgraph by taking the nodes belonging to two classes of interest (say $c_1$ and $c_2$). In equation 2 we substitute $p(c_1, c_2) = p_{bc}$, $p(c_1, c_1) = p_{wc}$, $p(\overline{c_1}) = p_1$ and $p(\overline{c_2}) = p_2$, the formula for LI becomes:

$$\text{LI}(G) = 2 - \frac{p_{bc} \log(p_{bc}) + p_{wc} \log(p_{wc})}{p_1 \log(p_1) + p_2 \log(p_2)} \tag{5}$$

where:

$$p_{bc} = \frac{e_{bc}}{e_{bc} + e_{wc}}, \quad p_{wc} = \frac{e_{wc}}{e_{bc} + e_{wc}}, \quad p_1 = \frac{2e_{bc} + e_{wc}}{2(e_{bc} + e_{wc})}, \quad p_2 = \frac{e_{wc}}{2(e_{bc} + e_{wc})}$$

In this equation, $e_{bc}$ represents the number of inter-class edges (between classes), and $e_{wc}$ represents the number of intra-class edges (within class). To understand how changes in the number of inter-class edges (denoted by $e_{bc}$) affect LI, we perform a differentiation of LI with respect to $e_{bc}$. Upon differentiating LI and evaluating it at ( $e_{bc}$ = 1.31167627 and $e_{wc}$ = 1 ), we find that the derivative is greater than 0. This positive derivative indicates that an increase in the number of inter-class edges increases LI. This result has significant implications for our understanding of graph structures and their label distributions. Specifically, it suggests that enhancing the connectivity between classes (increasing inter-class edges) can improve the informativeness of the labels. This improvement in LI can lead to better performance in tasks such as node classification, where the quality of label information is crucial.

### 4.2 Relationship between Label Informativeness and Accuracy with Mutual Information as a Mediator

In the field of Graph Neural Networks (GNNs), grasping the connection between label informativeness and accuracy is essential for enhancing model efficacy. When node labels contain highly informative data, GNN models are expected to generate more precise graph representations. This theorem has been developed to formalize this relationship, offering a mathematical framework to examine how label informativeness influences accuracy.

**Theorem 2.** *Let $I(Y, Z)$ be the Mutual Information between the node labels $Y$ and $Z$. $H(Y)$ be the entropy of the node labels, and $H(Y|Z)$ be the conditional entropy of the node labels. Then, the accuracy of the GNN model is directly proportional to the label informativeness.*

The proof is described in Appendix A.2.

## 5 EXPERIMENTS

### 5.1 DATASET

We have used standard datasets namely, Cora Sen et al. (2008), Twitter Mohammadrezaei et al. (2018), BlogCatalog Perozzi et al. (2014), Citeseer, PubMed, and Amazon McAuley & Leskovec (2013). Table 1 contains the statistics of the datasets used in this paper.

Cora, Citeseer, and PubMed are citation networks that are homophilous graph datasets. We have borrowed the long-tailed version of Cora and CiteSeer datasets used in Li et al. 2023. BlogCatalog and Twitter are social network datasets crawled from BlogCatalog and Twitter. Embedding vectors for each node for both of the graphs are obtained using Deepwalk. The Amazon graphs were constructed by connecting users based on shared product reviews (U-P-U), similar star ratings within a week (U-S-V), high mutual review text similarity (U-V-U), and using all three connections (All). All of these graphs along with the social network datasets are heterophilous.

Table 1: Data Statistics

| Dataset Name | Description | | | | |
| --- | --- | --- | --- | --- | --- |
| | Number of nodes | Number of edges | Average Degree | Number of classes | LI |
| Cora | 2708 | 5278 | 3.9 | 7 | 0.59 |
| Citeseer | 3327 | 4552 | 2.74 | 6 | 0.45 |
| PubMed | 19717 | 44324 | 4.5 | 3 | 0.41 |
| Twittter | 16587 | 393391 | 47.43 | 2 | 1.24E-05 |
| BlogCatalog | 10312 | 333983 | 64.78 | 38 | 0.01 |
| Amazon (U-P-U) | 11944 | 175608 | 29.41 | 2 | 0.004 |
| Amazon (U-S-U) | 11944 | 3566479 | 597.2 | 2 | 0.003 |
| Amazon (U-V-U) | 11944 | 1036737 | 173.6 | 2 | 0.005 |
| Amazon (All) | 11944 | 4398392 | 736.5 | 2 | 0.006 |

### 5.2 BASELINES

To evaluate LIMO's performance, we compared it against several state-of-the-art oversampling techniques, including Oversampling (OS), Re-weight (RW), SMOTE (SM), Embed-up (ES), and Graph SMOTE (GS). These methods represent various approaches to addressing class imbalance in graph data. Oversampling duplicates minority class samples, while Re-weight assigns higher weights to minority samples. SMOTE generates synthetic minority samples by interpolating between existing minority class samples in the feature space using

$$nn(v) = \arg \min_{u} \|f_u - f_v\|, \text{ s.t. } y_u = y_v \tag{6}$$

where where $nn(v)$ is the nearest neighbor of $v$ in the feature space, $f_u$ and $f_v$ are the features of u and v nodes, respectively, and $y_u$ and $y_v$ are the labels of u and v vertices. The features of the synthetic node, $v'$ are given by

$$f_{v'} = (1 - \delta) \times f_v + \delta \times f_{nn(v)} \tag{7}$$

where, $f_{nn(v)}$ is the feature vector of the nearest neighbor of $v$ and $\delta$ is a random variable taking value from 0 to 1. Embed-SMOTE is a variant of SMOTE adapted for deep learning, operating on the intermediate embedding layer of a GNN. Graph SMOTE is similar to SMOTE but generates synthetic nodes by interpolating in the embedding space and uses a neural network to predict edge existence.

### 5.3 RESULTS

LIMO consistently outperformed baseline methods across various imbalance ratios on most datasets. For homophilous (Cora LT) and heterophilous (Twitter) graphs, GNNs demonstrated significant performance improvements, especially for heterophilous datasets (figure 2). For instance, with an imbalance ratio of 0.4, GNNs achieved an $8.28\%$ performance boost compared to the best baseline

Table 2: Performance of baselines and LIMO with GraphSAGE for prediction on the Cora Long Tail dataset

| Imbalance ratio | Setting | LI | ACC (%) | AUC-ROC | F1-score |
|---|---|---|---|---|---|
| 0.6 | OS | 0.6035 ± 0.0000 | 86.20 ± 0.36 | 0.9762 ± 0.0006 | 0.8507 ± 0.0010 |
| | RW | 0.5904 ± 0.0000 | 85.93 ± 0.21 | 0.9763 ± 0.0006 | 0.8487 ± 0.0012 |
| | SM | 0.6035 ± 0.0000 | 85.73 ± 0.15 | 0.9761 ± 0.0004 | 0.8444 ± 0.0020 |
| | ES | 0.5904 ± 0.0000 | 86.07 ± 0.23 | 0.9764 ± 0.0006 | 0.8486 ± 0.0033 |
| | GS | 0.5904 ± 0.0000 | 85.37 ± 0.78 | 0.9750 ± 0.0022 | 0.8398 ± 0.0040 |
| | GraphSHA | 0.6465 ± 0.0001 | 87.50 ± 0.01 | 0.9843 ± 0.0000 | 0.8688 ± 0.0000 |
| | LIMO | **0.9693 ± 0.0000** | **92.67 ± 0.40** | **0.9898 ± 0.0000** | **0.9274 ± 0.0045** |
| 0.5 | OS | 0.6035 ± 0.0000 | 86.23 ± 0.42 | 0.9762 ± 0.0006 | 0.8512 ± 0.0008 |
| | RW | 0.5904 ± 0.0000 | 85.93 ± 0.21 | 0.9763 ± 0.0006 | 0.8487 ± 0.0012 |
| | SM | 0.6035 ± 0.0000 | 85.70 ± 0.10 | 0.9761 ± 0.0004 | 0.8441 ± 0.0017 |
| | ES | 0.5904 ± 0.0000 | 86.07 ± 0.23 | 0.9764 ± 0.0006 | 0.8486 ± 0.0033 |
| | GS | 0.5904 ± 0.0000 | 85.37 ± 0.78 | 0.9750 ± 0.0022 | 0.8398 ± 0.0040 |
| | GraphSHA | 0.6443 ± 0.0000 | 87.73 ± 0.00 | 0.9846 ± 0.0000 | 0.8686 ± 0.0000 |
| | LIMO | **0.9693 ± 0.0000** | **92.67 ± 0.40** | **0.9897 ± 0.0000** | **0.9274 ± 0.0045** |
| 0.4 | OS | 0.6035 ± 0.0000 | 86.10 ± 0.44 | 0.9757 ± 0.0009 | 0.8508 ± 0.0048 |
| | RW | 0.5904 ± 0.0000 | 85.93 ± 0.21 | 0.9763 ± 0.0006 | 0.8487 ± 0.0012 |
| | SM | 0.6035 ± 0.0000 | 85.70 ± 0.10 | 0.9761 ± 0.0004 | 0.8439 ± 0.0016 |
| | ES | 0.5904 ± 0.0000 | 86.07 ± 0.23 | 0.9764 ± 0.0006 | 0.8486 ± 0.0033 |
| | GS | 0.5904 ± 0.0000 | 85.37 ± 0.78 | 0.9750 ± 0.0022 | 0.8398 ± 0.0040 |
| | GraphSHA | 0.6464 ± 0.0000 | 87.27 ± 0.03 | 0.9842 ± 0.0000 | 0.8685 ± 0.0000 |
| | LIMO | **0.9693 ± 0.0000** | **92.67 ± 0.40** | **0.9898 ± 0.0000** | **0.9274 ± 0.0045** |
| 0.2 | OS | 0.6029 ± 0.0000 | 85.50 ± 0.20 | 0.9751 ± 0.0004 | 0.8407 ± 0.0019 |
| | RW | 0.5904 ± 0.0000 | 85.43 ± 0.57 | 0.9754 ± 0.0004 | 0.8395 ± 0.0059 |
| | SM | 0.6029 ± 0.0000 | 85.20 ± 0.52 | 0.9750 ± 0.0005 | 0.8354 ± 0.0060 |
| | ES | 0.5904 ± 0.0000 | 85.37 ± 0.15 | 0.9754 ± 0.0007 | 0.8391 ± 0.0013 |
| | GS | 0.5904 ± 0.0000 | 85.70 ± 0.70 | 0.9753 ± 0.0007 | 0.8418 ± 0.0100 |
| | GraphSHA | 0.6454 ± 0.0000 | 86.90 ± 0.00 | 0.9835 ± 0.0000 | 0.8658 ± 0.0000 |
| | LIMO | **0.9650 ± 0.0000** | **92.67 ± 0.23** | **0.9892 ± 0.0001** | **0.9269 ± 0.0028** |
| 0.1 | OS | 0.6004 ± 0.0000 | 84.23 ± 0.25 | 0.9696 ± 0.0008 | 0.8201 ± 0.0034 |
| | RW | 0.5904 ± 0.0000 | 84.27 ± 0.51 | 0.9693 ± 0.0005 | 0.8193 ± 0.0046 |
| | SM | 0.6004 ± 0.0000 | 84.13 ± 0.23 | 0.9689 ± 0.0001 | 0.8173 ± 0.0042 |
| | ES | 0.5904 ± 0.0000 | 83.93 ± 0.59 | 0.9694 ± 0.0011 | 0.8148 ± 0.0066 |
| | GS | 0.5904 ± 0.0000 | 83.67 ± 0.32 | 0.9685 ± 0.0027 | 0.8158 ± 0.0034 |
| | GraphSHA | 0.6284 ± 0.0002 | 85.67 ± 0.02 | 0.9783 ± 0.0000 | 0.8389 ± 0.0000 |
| | LIMO | **0.9485 ± 0.0000** | **92.03 ± 0.15** | **0.9871 ± 0.0004** | **0.9198 ± 0.0013** |

on Twitter. However, the impact of increasing LI was less pronounced on homophilous graphs, with a maximum improvement of $6.36\%$ on Cora LT compared to the best baseline.

We observe a diminishing effect of LI on GNN performance as the imbalance ratio decreases. This can be attributed to the limited number of potential edges that can be added to the graph with fewer minority class nodes. While LIMO can effectively increase LI, its impact is less significant in graphs with higher average degrees, as there are fewer opportunities for additional edge connections, as can be seen in the case of Amazon datasets (see table 1 and figure 7).

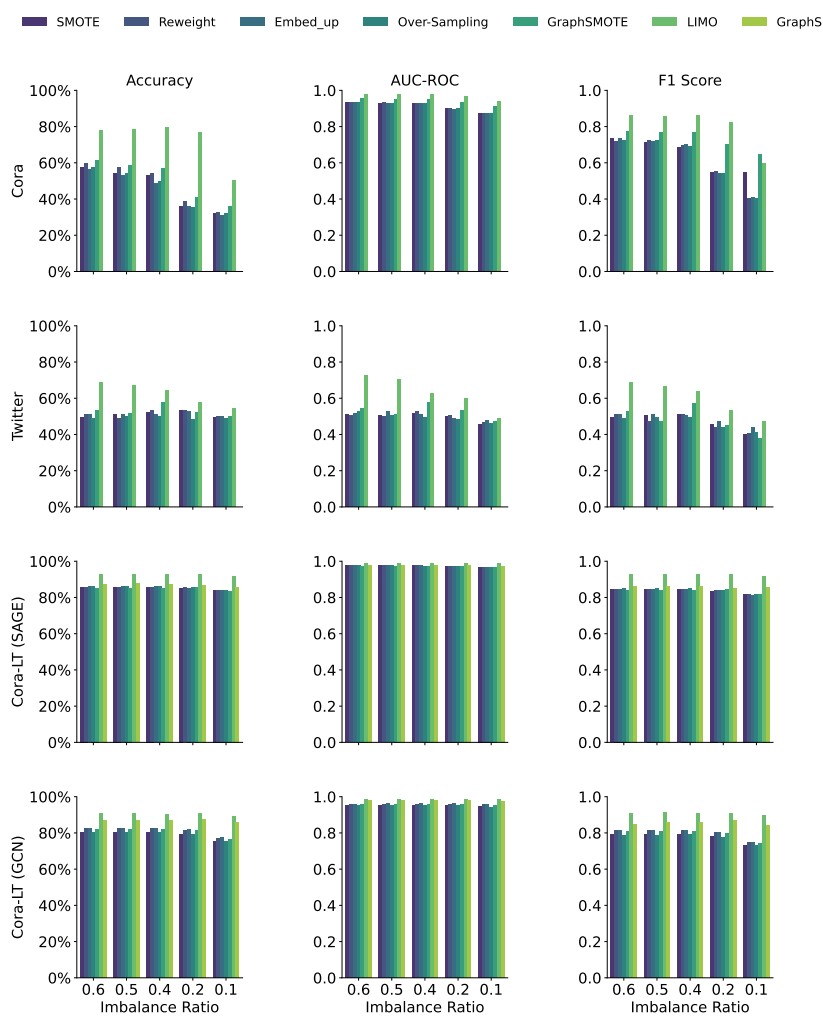

Figure 2: Performance of LIMO as compared to the baselines on Cora (Homophilous) and Twitter (Heterophilous) datasets

Due to the space limitation, we defer more experimental results on the other datasets (BlogCatalog, Citeseer, PubMed, Amazon (U-P-U), Amazon (U-S-U), Amazon (U-V-U), and Amazon (All)) in Appendix C.

## 5.4 PARAMETER SENSITIVITY

LIMO features two critical hyperparameters: node upscale and edge upscale. The node upscale parameter determines the multiplication factor for the existing nodes of the minority class to achieve a balanced dataset. Similarly, the edge upscale parameter specifies the multiplication factor for the total degrees of the minority nodes to generate edges in the balanced dataset. Our observations indicate a positive correlation between the performance and the number of edges added, as shown in figure 4. Especially for a graph with a low imbalance ratio, there is a positive correlation between

the multiple of synthetic minority nodes added and its LI and the performance of GNN trained on the graph (figure 3). This is confirmed by a weighted average of the Spearman rank-order correlation coefficient for LI and performance, which is $0.99 \pm 2 \times 10^{-12}$. This value is approximately equal to 1, which indicates that the LI and performance are directly proportional. We estimate these coefficients by calculating the Spearman coefficient between LI and Accuracy for each imbalance ratio for the data shown in figure 3. Then, we took the weighted average of all the Spearman coefficients for Cora using the inverse of the p-value as the weights. Same experiment on CiteSeer dataset also give similar stated in the appendix C.3

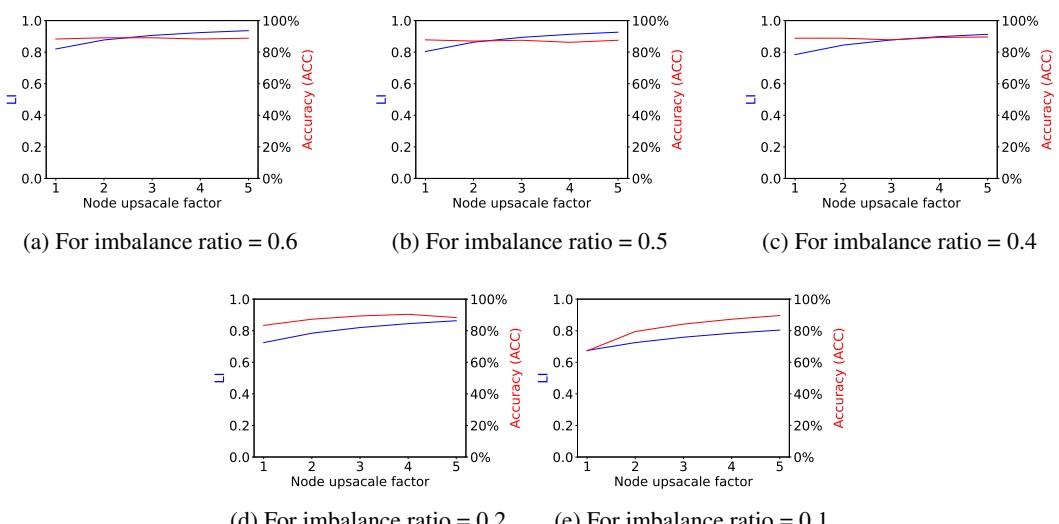

(a) For imbalance ratio = 0.6   (b) For imbalance ratio = 0.5   (c) For imbalance ratio = 0.4

(d) For imbalance ratio = 0.2   (e) For imbalance ratio = 0.1

Figure 3: The plots for LI of graph and performance of GNN on Cora dataset with different node upscale factors

## 5.5 ABLATION STUDY

We have claimed in this paper that the LI has a direct relationship with the performance of the GNN on the graph. To verify that, we designed a pair of experiments, one where we added different fractions of maximum edges that increase the LI of the resulting synthetic graph according to the algorithm 1 in increasing order, and we have noted the performance of the GNN trained on each of the synthetic datasets, the results of this experiment are shown in figure 4. In the other, we similarly added different fractions of maximum edges that decrease the LI of the resulting synthetic graph, whose results are as shown in the figure 5. These results are for the Cora dataset with different imbalance ratios for training. The results of both experiments agree with our claim of the existence of a positive correlation between LI and the performance of GNNs as the weighted average of the Spearman rank-order correlation coefficient for LI and performance is $0.8493 \pm 0.1283$, which is close to one. For GNN trained on Citeseer, the value is $0.75762 \pm 0.08054$. We estimate these coefficients by calculating the Spearman coefficient between LI and Accuracy for each imbalance ratio for the data shown in figure 4, 5, 9 and 10. Then, we took the weighted average of all the Spearman coefficients for Cora and Citeseer separately using the inverse of the p-value as the weights. Graphs for CiteSeer dataset can be found in the appendix C.4. The deviation from the trend in figure 4, for high fraction of edges added to increase LI, might be caused by over-fitting of the model on training dataset which thus even reduces the performance. In figure 5 we see that the Performance after some fraction of edges added (that reduce LI) the value of accuraacy does not show any significant correlation hence a high value of p-value for spearman coefficient like 0.13, 0.95, etc. which are all higher than 0.05. This might be because after a certain extent the accuracy of the model saturates and any further decrease in the LI due to addition of edges does not change the underlying graph structure.

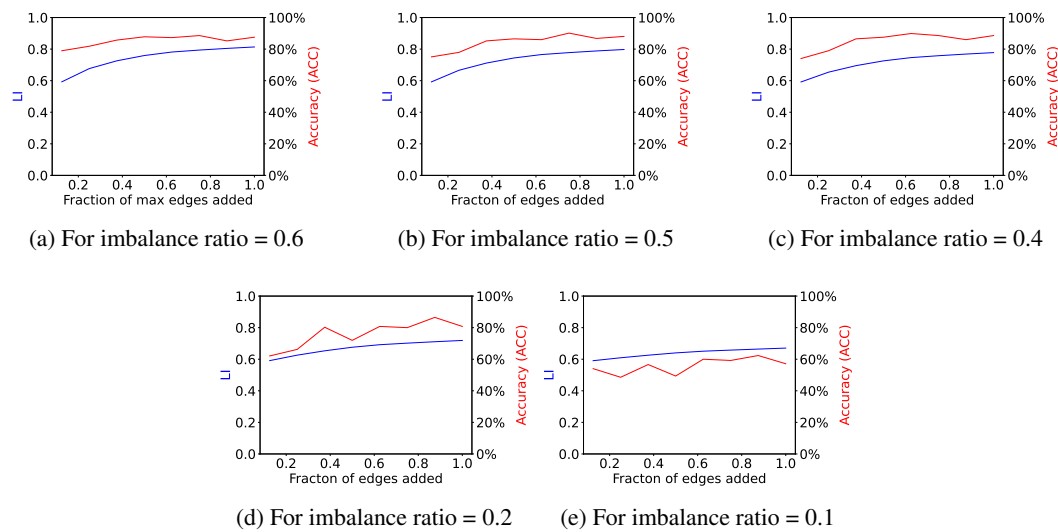

Figure 4: The plots for performance of GNN on Cora dataset with increase in LI

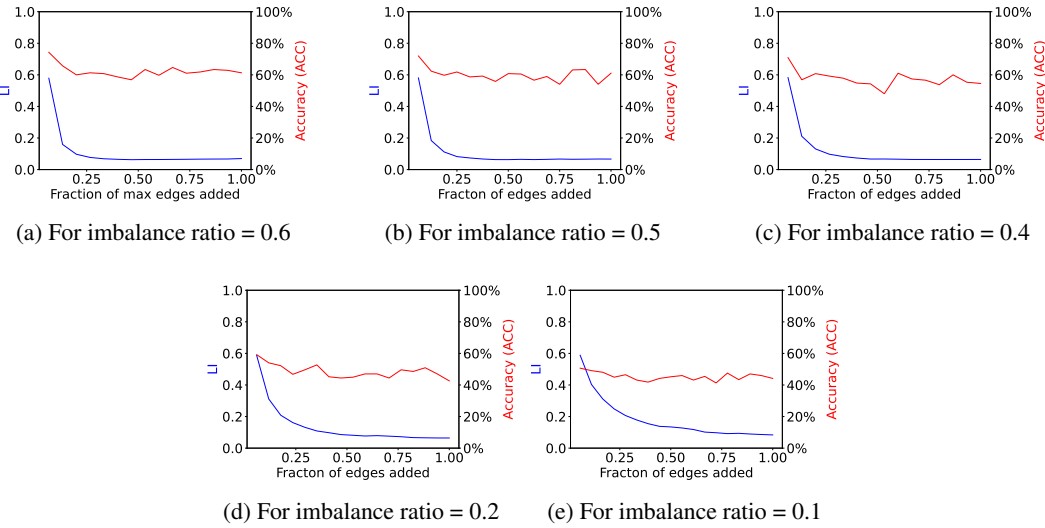

Figure 5: The plots for performance of GNN on Cora dataset with decrease in LI

## 6 RELATED WORKS

Data-level techniques on imbalanced graph data, such as oversampling and undersampling, aim to balance the dataset by increasing the number of instances in the minority class or reducing the number of instances in the majority class, respectively. However, undersampling may lead to the loss of potentially useful data (Khan & Chandra, 2024). Oversampling methods like SMOTE (Synthetic Minority Over-sampling Technique) Chawla et al., 2002 generate synthetic examples of the minority class by interpolating the feature space of the nodes to achieve a more balanced distribution. However, interpolation in the feature space can generate out-of-context synthetic nodes, which might lead to the biased learning of GNNs. Also, synthetic nodes borrow edges from their parent node, which might mislead the GNN. GraphSMOTE Zhao et al., 2021 mitigates this issue by using a GNN to predict the edges of synthetic nodes by learning from the graph itself. GraphSMOTE can be computationally intensive and complicated, which might lead to longer GNN training times.

ReNode Chen et al. (2021) addresses the problem of class imbalance from the perspective of topology imbalance and proposed a model-agnostic method designed to tackle the issue of topology

imbalance. It achieves this by adaptively re-weighting the influence of labeled nodes according to their relative positions to the class boundaries. $G^2$GNN Wang et al. (2022b) alleviates the graph imbalance issue by deriving extra supervision globally from neighboring graphs and locally from stochastic augmentations of graphs. The recent work HOVER Ashmore & Chen (2023) involves a simple yet effective edge removal method to mitigate heterophily and learn distinguishable node embeddings. These are then used to oversample minority bots to generate a balanced class distribution. FincGAN Hsu et al. (2024) employs a Generative Adversarial Network (GAN) to generate synthetic samples for minority classes, avoiding over-fitting issues common with traditional oversampling methods.

ImGAGN Qu et al., 2021 is an adversarial network-based architecture that adds a set of synthetic minority nodes to overcome the class imbalance. TAM Song et al., 2022 loss, which is topology-aware margin loss for class imbalanced node classification, performs well by comparing the connectivity pattern of each node with the class-averaged counterpart and adaptively adjusting the margin accordingly. mGNN Wang et al., 2022a mitigates the class imbalance by oversampling after performing the feature aggregation.

Recent studies such as Hsu et al. (2024) and Jing et al. (2024) tackle graph imbalance through oversampling methods. Hsu et al. (2024) utilizes GANs to create synthetic nodes and edges, but this approach is computationally expensive. Jing et al. (2024) employs dual-feature aggregation to address heterophily and conducts oversampling in the embedding space, avoiding edge synthesis. In contrast, the proposed LIMO directly increases a graph property LI through strategic node and edge augmentation. This increases model performance and improves minority class representation.

In the field of graph-based learning, several innovative approaches have emerged to address class imbalance. These include Park et al. (2022), which generates ego networks for minority class nodes while maintaining structural consistency and employing saliency-based node mixing to avoid introducing class-specific features. Another method, Song et al. (2022), implements a topology-aware margin loss to enhance the separation of minority class nodes while preserving graph structure. Additionally, Li et al. (2023) creates more challenging samples for underrepresented classes, thereby enhancing training effectiveness in imbalanced scenarios. While these techniques offer innovative solutions, LIMO sets itself apart by directly utilizing Label Informativeness (LI) to guide both node and edge augmentation. This approach results in balanced graph representations that are specifically optimized for downstream GNN tasks.

## 7 CONCLUSION

In this work, we introduced Label Informativeness-based Minority Oversampling (LIMO), a novel approach to addressing class imbalance in graph-structured data. ~~By augmenting edges in a manner that maximizes Label Informativeness (LI), LIMO strategically oversamples minority class nodes without significantly inflating the dataset.~~ Our theoretical analysis revealed that the performance of Graph Neural Networks (GNNs) is strongly correlated with label informativeness, with mutual information acting as a key intermediary. The analysis of the derivative of LI further provided a deeper understanding of the impact of inter-class edges on informativeness. Experimental results on various benchmark datasets, both homophilous and heterophilous, demonstrated that LIMO substantially improves node classification accuracy, particularly in heterophilous settings where class imbalance is more pronounced.

Despite its strengths, LIMO has certain limitations. First, while the algorithm balances the dataset without inflating it excessively, there is still an inherent computational cost associated with generating and evaluating new edges. Second, LIMO's effectiveness depends on the accuracy of label informativeness estimation, which may be less reliable in graphs where the node labels exhibit low correlation with their neighbors. Overcoming these limitations remains an open area for future investigation.

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

# A  LABEL INFORMATIVENESS AND ACCURACY

## A.1  CONDITIONS FOR IMPROVING LABEL INFORMATIVENESS THROUGH EDGE ADDITION

Proof of theorem 1.

*Proof.* To prove this theorem, we need to show that the derivative of LI with respect to $e_{bc}$ is positive when the condition $e_{bc} \cdot 1.31167627 > e_{wc}$ is satisfied.

First, let's rewrite the equation for LI in terms of $e_{bc}$ and $e_{wc}$:

$$LI = 2 - \frac{\frac{e_{bc}}{e_{bc}+e_{wc}} \log\left(\frac{e_{bc}}{e_{bc}+e_{wc}}\right) + \frac{e_{wc}}{e_{bc}+e_{wc}} \log\left(\frac{e_{wc}}{e_{bc}+e_{wc}}\right)}{\frac{2e_{bc}+e_{wc}}{2(e_{bc}+e_{wc})} \log\left(\frac{2e_{bc}+e_{wc}}{2(e_{bc}+e_{wc})}\right) + \frac{e_{wc}}{2(e_{bc}+e_{wc})} \log\left(\frac{e_{wc}}{2(e_{bc}+e_{wc})}\right)}$$

Now, let's compute the derivative of LI with respect to $e_{bc}$:

$$\frac{dLI}{de_{bc}} = \left( \frac{-1}{\frac{2e_{bc}+e_{wc}}{2e_{bc}+2e_{wc}} \log\left(\frac{2e_{bc}+e_{wc}}{2e_{bc}+2e_{wc}}\right) + \frac{e_{wc}}{2e_{bc}+2e_{wc}} \log\left(\frac{e_{wc}}{2e_{bc}+2e_{wc}}\right)} \right) \left( \frac{e_{wc}}{(e_{bc}+e_{wc})^2} \right) \log\left(\frac{e_{bc}}{e_{wc}}\right)$$

$$+ \left( \frac{\frac{e_{bc}}{e_{bc}+e_{wc}} \log\left(\frac{e_{bc}}{e_{bc}+e_{wc}}\right) + \frac{e_{wc}}{e_{bc}+e_{wc}} \log\left(\frac{e_{wc}}{e_{bc}+e_{wc}}\right)}{\left(\frac{2e_{bc}+e_{wc}}{2e_{bc}+2e_{wc}} \log\left(\frac{2e_{bc}+e_{wc}}{2e_{bc}+2e_{wc}}\right) + \frac{e_{wc}}{2e_{bc}+2e_{wc}} \log\left(\frac{e_{wc}}{2e_{bc}+2e_{wc}}\right)\right)^2} \right) \left( \frac{2e_{wc}}{(2e_{bc}+2e_{wc})^2} \right) \log\left(\frac{2e_{bc}+e_{wc}}{e_{wc}}\right)$$

While fixing $e_{wc}$ to be 1 and changing $e_{bc}$ from 0 to 100 in the steps of 0.00000001 and noting the corresponding value of $\frac{dLI}{de_{bc}}$ at each step, we found that for $e_{bc} > \frac{1}{t}$ where t= 1.31167628 , we get:

$$\frac{dLI}{de_{bc}} > 0$$

Similarly differentiating LI with respect to $e_{wc}$ and fixing $e_{bc}$ to be 1 and changing $e_{wc}$ from 0 to 100 in the steps of 0.00000001 and noting the corresponding value of $\frac{dLI}{de_{wc}}$ at we found that for $e_{wc} < t$ we get:

$$\frac{dLI}{de_{wc}} > 0$$

Since $\frac{dLI}{de_{bc}}$ and $\frac{dLI}{de_{wc}}$ is positive, increasing $e_{bc}$ beyond $\frac{1}{t}$ and $e_{wc}$ upto $t$ will increase the Label Informativeness (LI).

The value of $t$ obtained here is calculated keeping in mind the number of edges between classes and within classes. But this value can be separately calculated for edges within each class and edges between multiple classes. However, this would be computationally expensive and unnecessary as the experiment shows that this value is the same for all graphs and the class of interest. Thus this approach was not explained in this paper.

For more information on the calculation of $t$ refer to the file named experiment.ipynb in the repository https://anonymous.4open.science/r/limo-12CC/

$\square$

## A.2 RELATIONSHIP BETWEEN LABEL INFORMATIVENESS AND ACCURACY WITH MUTUAL INFORMATION AS A MEDIATOR

The proof of theorem 2.

*Proof.* The Mutual Information between the node labels $Y$ and $Z$ is defined as Platonov et al. (2024):

$$I(Y, Z) = H(Y) - H(Y|Z)$$

The accuracy of the GNN model is related to the error rate as Google (2024):

$$\text{Accuracy} = 1 - \text{Error Rate}$$

The error rate is defined as:

$$\text{Error Rate} = \frac{H(Y|Z)}{H(Y)}$$

Substituting the error rate into the Mutual Information equation, we get:

$$I(Y, Z) = H(Y) - H(Y) \times \text{Error Rate}$$

Simplifying the equation, we get:

$$I(Y, Z) = H(Y) \times (1 - \text{Error Rate})$$

Substituting the accuracy equation, we get:

$$I(Y, Z) = H(Y) \times \text{Accuracy}$$

We also know from Platonov et al. (2024) that:

$$\text{LI}(G) = \frac{I(Y, Z)}{H(Y)}$$

Therefore, the label informativeness $\text{LI}(G)$ is directly proportional to the accuracy of the GNN model.

$\square$

## A.3 COMPLEXITY ANALYSIS

In this section, we note the computational cost of the LIMO algorithm, accounting for both node and edge additions.

SPACE COMPLEXITY

- LIMO (Node Addition): The number of nodes added by LIMO is equal to the number of minority class nodes, denoted as $n_{\text{minority}}$. Thus, the space required for additional nodes is $O(n_{\text{minority}})$.
- LIMO (Edge Addition): The number of edges added depends on the original *LI* value and the adjacency matrix of the graph. This number, denoted as $n_{\text{edges}}$, can vary across datasets. Hence, the space complexity for edges is $O(n_{\text{edges}})$.

TIME COMPLEXITY

- LIMO (Node Addition) For each new node added by SMOTE, we need to find the nearest nodes in the feature space across all the nodes in the graph. Finding the nearest nodes requires $O(n)$, where $n$ is the total number of nodes in the graph. Since we add $n_{\text{minority}}$ nodes, the total time complexity for SMOTE is: $O(n_{\text{minority}} \cdot n)$

- **LIMO Edge Addition** For every edge added, calculating $pc_1$ and $pc_2$ (graph-specific properties) requires $O(n^2)$. This is calculated for all the classes: $O(C \cdot n^2)$, where $C$ is the number of classes.

  For each newly generated node, which is $n_{\text{minority}}$ in number, we have to compare edge probability $pc_1$ and $pc_2$ across all classes. time complexity for this is $O(C \cdot n_{\text{minority}})$. Since $O(C \cdot n_{\text{minority}})$ is smaller than $O(C \cdot n^2)$, it can be ignored.

Overall time complexity by combining the above: Total Time Complexity $= O(n_{\text{minority}} \cdot n) + O(C \cdot n^2)$

## B    EVALUATION METRICS

### B.1    EVALUATION METRICS

We have used Accuracy, AUC-ROC, and F1 scores to judge the performance of the GNN on imbalanced datasets. The details of these metrics are as follows:

Accuracy

Accuracy measures the proportion of correct predictions made by the model out of all predictions made. It is calculated using the formula:

$$\text{Accuracy} = \frac{\text{Number of Correct Predictions}}{\text{Total Number of Predictions}}$$

While this is a good metric to infer the overall performance of a model, it fails to tell the whole story when there is a stark imbalance in the class distribution of the nodes. Even if all the minority is classified as majority class the accuracy score will be high. It also fails when the detection of one class correctly is more important than the other classes.

Area under the receiver operating characteristic curve (AUC-ROC)

It is a performance measurement for a classification problem defined as the area under the true positive rate versus the false positive rate plot, ranging from 0 to 1. The true positive rate, also known as sensitivity or recall, is the ratio of the correctly predicted positives to the sum of the correctly predicted positives and incorrectly predicted negatives. The false positive rate is the ratio of incorrectly predicted positives to the sum of incorrectly predicted positives and correctly predicted negatives.

F1-Score

The metric is defined as the harmonic mean of precision and recall, ranging from 0 to 1. Precision is the ratio of correctly predicted positives to all positives. Recall is the same as was defined for AUC-ROC.

$$\text{F1-Score} = 2 \times \frac{\text{Precision} \times \text{Recall}}{\text{Precision} + \text{Recall}}$$

### B.2    BASELINES

To evaluate LIMO's performance, we compared it against several state-of-the-art oversampling techniques, including Oversampling (OS), Re-weight (RW), SMOTE (SM), Embed-up (ES), and Graph SMOTE (GS). A brief description of the baselines is given below:

1. Over-sampling (OS) is a classical approach for imbalanced learning problems by repeating samples from minority classes. We implement it in the raw input space by duplicating $n_s$ minority nodes along their edges. In each training iteration, V is over-sampled to contain $n + n_s$ nodes, and $A \in \mathbf{R}^{(n+n_s) \times (n+n_s)}$. Zhao et al. (2021)

2. Re-weightYuan & Ma (2012) (RW) is a cost-sensitive approach that gives class-specific loss weight. In particular, it assigns higher loss weights to minority samples to alleviate the issue of majority classes dominating the loss function.

3. SMOTE (Synthetic minority over-sampling technique)Chawla et al. (2002) (SM) in the feature space is the interpolation of the synthetic data point between the target node and the node that is nearest to it in the feature space, nn(v) as given by

$$nn(v) = \arg\min_u \|f_u - f_v\|, \text{ s.t. } Y_u = Y_v$$

where $f_u$ and $f_v$ are the features of u and v nodes, respectively, and $Y_v$ are the labels of u and v vertices. The features of the synthetic node are given by

$$f_{v'} = (1 - \delta) \times f_v + \delta \times f_n n(v)$$

4. Embed-SMOTE (ES) Ando & Huang (2017) A variant of SMOTE adapted for deep learning, designed to oversample data within the intermediate embedding layer of a GNN. This approach eliminates the need for edge generation by operating directly on the learned representations.

5. GraphSMOTE (GS) Zhao et al. (2021) generates synthetic nodes similar to smote, but here, it interpolates in the embedding space to create the embedding of the synthetic node. There is an edge generator that learns using a neural network whether there exists an edge between given nodes. A GNN then does the classification; some versions also use the loss in the classification to train the embedding and the edge generator.

### B.3 HARDWARE AND SOFTWARE SPECIFICATIONS

We experiment on each dataset using a standard split of 20 nodes for training, 25 for validation, and 55 for testing in the majority class. For minority classes with an imbalance ratio $i \in [0, 1]$, we sampled $20 \times i$ nodes. When the minority class had fewer than three nodes, we allocated one node each for training, validation, and testing. To evaluate LIMO, we compared it with several baselines: Over-sampling (OS), Reweight (RW), SMOTE (SM), Embed-SMOTE (ES), and Graph SMOTE (GS). All experiments were performed on NVIDIA GeForce RTX 3090, A100-SXM4-80GB, and RTX 6000 Ada Generation GPUs in Python using PyTorch and PyG. We employed GraphSAGE as the GNN architecture for training on the balanced datasets created by LIMO and the baselines. We conducted experiments with three random seeds (10, 20, 30) to mitigate randomness and averaged the results. The GraphSAGE model used two layers with a linear layer output dimension of 64 for both layers. ReLU activation was employed, and we used Adam optimizer for training with a learning rate of 0.001. We either terminate the training after 5000 epochs or when validation performance plateaued.

## C ADDITIONAL RESULTS

### C.1 HOMPHILOUS DATA

This section contains the results for the performance of GNNs (GraphSAGE and GCN) on some more homophilous datasets (CiteSeer, CiteSeer Long Tail, and PubMed).

Table 3: Performance of baselines and LIMO on the Cora dataset

| Imbalance ratio | Setting | LI | ACC (%) | AUC-ROC | F1-score |
|---|---|---|---|---|---|
| 0.6 | OS | 0.5920 ± 0.0031 | 73.07 ± 3.65 | 0.9331 ± 0.0144 | 0.7281 ± 0.0384 |
| | RW | 0.5904 ± 0.0000 | 72.47 ± 3.61 | 0.9358 ± 0.0123 | 0.7198 ± 0.0386 |
| | SM | 0.5920 ± 0.0031 | 73.77 ± 3.49 | 0.9355 ± 0.0136 | 0.7353 ± 0.0362 |
| | ES | 0.5904 ± 0.0000 | 74.29 ± 3.51 | 0.9363 ± 0.0121 | 0.7395 ± 0.0362 |
| | GS | 0.5904 ± 0.0000 | 77.83 ± 2.21 | 0.9550 ± 0.0076 | 0.7760 ± 0.0228 |
| | LIMO | **0.8200 ± 0.0000** | **86.92 ± 1.20** | **0.9820 ± 0.0071** | **0.8657 ± 0.0127** |
| 0.5 | OS | 0.5919 ± 0.0027 | 72.73 ± 3.06 | 0.9321 ± 0.0142 | 0.7232 ± 0.0307 |
| | RW | 0.5904 ± 0.0000 | 73.16 ± 3.38 | 0.9337 ± 0.0136 | 0.7283 ± 0.0351 |
| | SM | 0.5919 ± 0.0027 | 72.03 ± 3.64 | 0.9328 ± 0.0153 | 0.7157 ± 0.0383 |
| | ES | 0.5904 ± 0.0000 | 72.73 ± 3.57 | 0.9327 ± 0.0143 | 0.7226 ± 0.0371 |
| | GS | 0.5904 ± 0.0000 | 77.40 ± 1.58 | 0.9531 ± 0.0038 | 0.7717 ± 0.0159 |
| | LIMO | **0.8039 ± 0.0000** | **86.06 ± 2.45** | **0.9801 ± 0.0084** | **0.8573 ± 0.0263** |
| 0.4 | OS | 0.5917 ± 0.0026 | 70.22 ± 3.65 | 0.9293 ± 0.0142 | 0.6947 ± 0.0427 |
| | RW | 0.5904 ± 0.0000 | 70.65 ± 4.18 | 0.9300 ± 0.0145 | 0.6994 ± 0.0486 |
| | SM | 0.5917 ± 0.0026 | 69.35 ± 3.00 | 0.9278 ± 0.0121 | 0.6861 ± 0.0374 |
| | ES | 0.5904 ± 0.0000 | 70.74 ± 3.77 | 0.9296 ± 0.0144 | 0.7013 ± 0.0425 |
| | GS | 0.5904 ± 0.0000 | 77.23 ± 1.69 | 0.9549 ± 0.0065 | 0.7699 ± 0.0180 |
| | LIMO | **0.7841 ± 0.0000** | **86.66 ± 1.92** | **0.9803 ± 0.0084** | **0.8647 ± 0.0192** |
| 0.2 | OS | 0.5906 ± 0.0014 | 58.09 ± 3.79 | 0.9003 ± 0.0187 | 0.5458 ± 0.0464 |
| | RW | 0.5904 ± 0.0000 | 59.22 ± 3.41 | 0.9004 ± 0.0184 | 0.5570 ± 0.0433 |
| | SM | 0.5906 ± 0.0014 | 58.61 ± 3.79 | 0.9007 ± 0.0201 | 0.5498 ± 0.0457 |
| | ES | 0.5904 ± 0.0000 | 58.18 ± 3.60 | 0.8979 ± 0.0203 | 0.5410 ± 0.0510 |
| | GS | 0.5904 ± 0.0000 | 71.34 ± 5.46 | 0.9361 ± 0.0161 | 0.7019 ± 0.0654 |
| | LIMO | **0.7246 ± 0.0000** | **82.25 ± 1.33** | **0.9678 ± 0.0104** | **0.8238 ± 0.0135** |
| 0.1 | OS | 0.5917 ± 0.0002 | 48.92 ± 3.43 | 0.8742 ± 0.0290 | 0.4043 ± 0.0404 |
| | RW | 0.5904 ± 0.0000 | 49.26 ± 2.34 | 0.8766 ± 0.0271 | 0.4055 ± 0.0251 |
| | SM | 0.5917 ± 0.0002 | 49.35 ± 3.78 | 0.8758 ± 0.0287 | 0.4101 ± 0.0433 |
| | ES | 0.5904 ± 0.0000 | 49.70 ± 1.73 | 0.8762 ± 0.0270 | 0.4107 ± 0.0158 |
| | GS | 0.5904 ± 0.0000 | 66.76 ± 6.56 | 0.9123 ± 0.0395 | 0.6503 ± 0.0787 |
| | LIMO | **0.6748 ± 0.0000** | **62.08 ± 4.43** | **0.9402 ± 0.0167** | **0.6003 ± 0.0498** |

Table 4: Performance of baselines and LIMO with GCN for prediction on the Cora Long Tail dataset

| Imbalance ratio | Setting | LI | ACC (%) | AUC-ROC | F1-score |
|---|---|---|---|---|---|
| 0.6 | OS | 0.6035 ± 0.0000 | 80.10% ± 0.20% | 0.9519 ± 0.0013 | 0.7893 ± 0.0036 |
| | RW | 0.5904 ± 0.0000 | 82.57% ± 0.21% | 0.9602 ± 0.0007 | 0.8156 ± 0.0030 |
| | SM | 0.6035 ± 0.0000 | 80.30% ± 0.10% | 0.9510 ± 0.0004 | 0.7915 ± 0.0030 |
| | ES | 0.5904 ± 0.0000 | 82.43% ± 0.29% | 0.9607 ± 0.0006 | 0.8130 ± 0.0029 |
| | GS | 0.5943 ± 0.0022 | 81.97% ± 1.00% | 0.9587 ± 0.0015 | 0.8083 ± 0.0124 |
| | GraphSHA | 0.6815 ± 0.0000 | 87.10 ± 0.01 | 0.9820 ± 0.0000 | 0.8570 ± 0.0000 |
| | **LIMO** | **0.9693 ± 0.0000** | **90.57% ± 0.42%** | **0.9871 ± 0.0001** | **0.9095 ± 0.0036** |
| 0.5 | OS | 0.6035 ± 0.0000 | 80.07% ± 0.06% | 0.9520 ± 0.0012 | 0.7892 ± 0.0017 |
| | RW | 0.5904 ± 0.0000 | 82.57% ± 0.21% | 0.9602 ± 0.0007 | 0.8156 ± 0.0030 |
| | SM | 0.6035 ± 0.0000 | 80.30% ± 0.10% | 0.9508 ± 0.0004 | 0.7912 ± 0.0027 |
| | ES | 0.5904 ± 0.0000 | 82.60% ± 0.00% | 0.9611 ± 0.0000 | 0.8150 ± 0.0000 |
| | GS | 0.5943 ± 0.0022 | 81.97% ± 1.00% | 0.9587 ± 0.0015 | 0.8083 ± 0.0124 |
| | GraphSHA | 0.6820 ± 0.0001 | 87.20 ± 0.01 | 0.9820 ± 0.0000 | 0.8577 ± 0.0000 |
| | **LIMO** | **0.9693 ± 0.0000** | **90.90% ± 0.82%** | **0.9874 ± 0.0001** | **0.9138 ± 0.0064** |
| 0.4 | OS | 0.6035 ± 0.0000 | 80.07% ± 0.15% | 0.9522 ± 0.0010 | 0.7904 ± 0.0045 |
| | RW | 0.5904 ± 0.0000 | 82.57% ± 0.21% | 0.9602 ± 0.0007 | 0.8156 ± 0.0030 |
| | SM | 0.6035 ± 0.0000 | 80.27% ± 0.15% | 0.9510 ± 0.0005 | 0.7911 ± 0.0035 |
| | ES | 0.5904 ± 0.0000 | 82.60% ± 0.00% | 0.9611 ± 0.0000 | 0.8150 ± 0.0000 |
| | GS | 0.5943 ± 0.0022 | 81.97% ± 1.00% | 0.9587 ± 0.0015 | 0.8083 ± 0.0124 |
| | GraphSHA | 0.6821 ± 0.0000 | 87.07 ± 0.01 | 0.9820 ± 0.0000 | 0.8575 ± 0.0001 |
| | **LIMO** | **0.9693 ± 0.0000** | **90.23% ± 0.55%** | **0.9871 ± 0.0002** | **0.9077 ± 0.0043** |
| 0.2 | OS | 0.6029 ± 0.0000 | 79.10% ± 0.35% | 0.9532 ± 0.0005 | 0.7752 ± 0.0047 |
| | RW | 0.5904 ± 0.0000 | 81.47% ± 0.58% | 0.9599 ± 0.0013 | 0.8023 ± 0.0053 |
| | SM | 0.6029 ± 0.0000 | 79.47% ± 0.12% | 0.9532 ± 0.0017 | 0.7795 ± 0.0022 |
| | ES | 0.5904 ± 0.0000 | 81.80% ± 0.56% | 0.9613 ± 0.0006 | 0.8053 ± 0.0073 |
| | GS | 0.5928 ± 0.0010 | 81.43% ± 0.32% | 0.9580 ± 0.0004 | 0.7988 ± 0.0054 |
| | GraphSHA | 0.6796 ± 0.0000 | 87.37 ± 0.00 | 0.9814 ± 0.0000 | 0.8596 ± 0.0001 |
| | **LIMO** | **0.9650 ± 0.0000** | **90.60% ± 0.26%** | **0.9871 ± 0.0002** | **0.9096 ± 0.0015** |
| 0.1 | OS | 0.6004 ± 0.0000 | 75.60% ± 0.70% | 0.9439 ± 0.0012 | 0.7307 ± 0.0095 |
| | RW | 0.5904 ± 0.0000 | 77.03% ± 0.15% | 0.9561 ± 0.0014 | 0.7477 ± 0.0034 |
| | SM | 0.6004 ± 0.0000 | 75.53% ± 0.38% | 0.9450 ± 0.0025 | 0.7311 ± 0.0053 |
| | ES | 0.5904 ± 0.0000 | 77.50% ± 0.36% | 0.9569 ± 0.0009 | 0.7504 ± 0.0069 |
| | GS | 0.5929 ± 0.0019 | 76.57% ± 0.67% | 0.9530 ± 0.0026 | 0.7422 ± 0.0083 |
| | GraphSHA | 0.6631 ± 0.0000 | 85.80 ± 0.02 | 0.9733 ± 0.0000 | 0.8438 ± 0.0001 |
| | **LIMO** | **0.9485 ± 0.0000** | **89.37% ± 0.45%** | **0.9858 ± 0.0004** | **0.8952 ± 0.0047** |

Table 5: Table for the performance of baselines and LIMO on the CiteSeer dataset with node classification using GraphSAGE

| Imbalance ratio | Setting | LI | ACC (%) | AUC-ROC | F1-score |
|---|---|---|---|---|---|
| 0.6 | OS | 0.452778 ± 0.002726 | 57.47 ± 6.24 | 0.8558 ± 0.0225 | 0.5750 ± 0.0591 |
| | RW | 0.452147 ± 0.002403 | 60.10 ± 4.04 | 0.8662 ± 0.0093 | 0.5983 ± 0.0390 |
| | SM | 0.452778 ± 0.002726 | 57.88 ± 4.71 | 0.8565 ± 0.0247 | 0.5780 ± 0.0457 |
| | ES | 0.450760 ± 0.000000 | 56.36 ± 5.78 | 0.8497 ± 0.0239 | 0.5636 ± 0.0551 |
| | GS | 0.450760 ± 0.000000 | 61.52 ± 5.16 | 0.8776 ± 0.0221 | 0.6141 ± 0.0555 |
| | **LIMO** | **0.849402 ± 0.000000** | **77.98 ± 2.61** | **0.9596 ± 0.0101** | **0.7663 ± 0.0307** |
| 0.5 | OS | 0.452367 ± 0.002494 | 54.34 ± 6.28 | 0.8467 ± 0.0223 | 0.5390 ± 0.0643 |
| | RW | 0.452017 ± 0.002178 | 57.47 ± 4.70 | 0.8576 ± 0.0153 | 0.5720 ± 0.0535 |
| | SM | 0.452367 ± 0.002494 | 54.34 ± 7.38 | 0.8457 ± 0.0279 | 0.5412 ± 0.0747 |
| | ES | 0.450760 ± 0.000000 | 53.43 ± 6.12 | 0.8420 ± 0.0241 | 0.5300 ± 0.0643 |
| | GS | 0.450760 ± 0.000000 | 58.79 ± 3.72 | 0.8661 ± 0.0162 | 0.5859 ± 0.0356 |
| | **LIMO** | **0.831329 ± 0.000000** | **78.39 ± 3.34** | **0.9562 ± 0.0137** | **0.7708 ± 0.0381** |
| 0.4 | OS | 0.452246 ± 0.000876 | 49.90 ± 6.97 | 0.8360 ± 0.0306 | 0.4899 ± 0.0779 |
| | RW | 0.451579 ± 0.001419 | 54.24 ± 4.27 | 0.8518 ± 0.0041 | 0.5398 ± 0.0552 |
| | SM | 0.452814 ± 0.000698 | 53.03 ± 8.92 | 0.8350 ± 0.0356 | 0.5317 ± 0.0977 |
| | ES | 0.450760 ± 0.000000 | 48.58 ± 7.43 | 0.8292 ± 0.0384 | 0.4767 ± 0.0811 |
| | GS | 0.450760 ± 0.000000 | 57.17 ± 6.48 | 0.8623 ± 0.0219 | 0.5715 ± 0.0651 |
| | **LIMO** | **0.807730 ± 0.000000** | **79.80 ± 1.95** | **0.9618 ± 0.0084** | **0.7886 ± 0.0204** |
| 0.2 | OS | 0.451628 ± 0.000179 | 35.45 ± 4.01 | 0.7673 ± 0.0401 | 0.2884 ± 0.0461 |
| | RW | 0.451069 ± 0.000536 | 38.79 ± 1.32 | 0.7796 ± 0.0235 | 0.3329 ± 0.0124 |
| | SM | 0.451628 ± 0.000179 | 36.26 ± 4.43 | 0.7781 ± 0.0364 | 0.2960 ± 0.0523 |
| | ES | 0.450760 ± 0.000000 | 36.36 ± 2.48 | 0.7628 ± 0.0322 | 0.3057 ± 0.0270 |
| | GS | 0.450760 ± 0.000000 | 41.01 ± 6.13 | 0.7959 ± 0.0601 | 0.3676 ± 0.0666 |
| | **LIMO** | **0.727124 ± 0.000000** | **76.87 ± 4.66** | **0.9423 ± 0.0121** | **0.7689 ± 0.0473** |
| 0.1 | OS | 0.451415 ± 0.000058 | 32.53 ± 3.24 | 0.7417 ± 0.0402 | 0.2259 ± 0.0313 |
| | RW | 0.451000 ± 0.000417 | 32.93 ± 2.81 | 0.7428 ± 0.0382 | 0.2401 ± 0.0102 |
| | SM | 0.451415 ± 0.000058 | 32.33 ± 4.11 | 0.7447 ± 0.0394 | 0.2268 ± 0.0367 |
| | ES | 0.450760 ± 0.000000 | 31.31 ± 2.82 | 0.7196 ± 0.0391 | 0.2256 ± 0.0207 |
| | **LIMO** | **0.644973 ± 0.000000** | **50.60 ± 8.20** | **0.8858 ± 0.0168** | **0.5040 ± 0.0863** |

Table 6: Performance of baselines and LIMO on the CiteSeer Long Tail dataset with node classification using GraphSAGE

| Imbalance ratio | Setting | LI | ACC (%) | AUC-ROC | F1-score |
|---|---|---|---|---|---|
| 0.6 | OS | 0.478898 ± 0.000000 | 78.53 ± 0.51 | 0.9343 ± 0.0004 | 0.7497 ± 0.0083 |
| | RW | 0.450760 ± 0.000000 | 78.70 ± 0.61 | 0.9331 ± 0.0004 | 0.7496 ± 0.0095 |
| | SM | 0.478898 ± 0.000000 | 78.40 ± 0.35 | 0.9340 ± 0.0005 | 0.7462 ± 0.0044 |
| | ES | 0.450760 ± 0.000000 | 76.13 ± 0.83 | 0.9297 ± 0.0010 | 0.7138 ± 0.0067 |
| | GS | 0.450760 ± 0.000000 | 76.97 ± 2.50 | 0.9283 ± 0.00 | 0.7267 ± 0.0418 |
| | GraphSHA | 0.698450 ± 0.000139 | 77.53 ± 0.00 | 0.9365 ± 0.0000 | 0.7398 ± 0.0000 |
| | LIMO | **0.990356 ± 0.000000** | **89.40 ± 0.66** | **0.9751 ± 0.0035** | **0.8559 ± 0.0106** |
| 0.5 | OS | 0.477518 ± 0.000000 | 78.30 ± 0.30 | 0.9332 ± 0.0005 | 0.7502 ± 0.0068 |
| | RW | 0.450760 ± 0.000000 | 78.1 ± 0.15 | 0.9322 ± 0.0000 | 0.7482 ± 0.0045 |
| | SM | 0.477518 ± 0.000000 | 78.03 ± 0.45 | 0.9328 ± 0.0002 | 0.7451 ± 0.0042 |
| | ES | 0.450760 ± 0.000000 | 78.20 ± 0.35 | 0.9325 ± 0.0000 | 0.7480 ± 0.0058 |
| | GS | 0.450760 ± 0.000000 | 77.93 ± 0.38 | 0.9303 ± 0.0010 | 0.7500 ± 0.0036 |
| | GraphSHA | 0.706850 ± 0.000257 | 77.17 ± 0.01 | 0.9364 ± 0.0000 | 0.7327 ± 0.0000 |
| | LIMO | **0.989453 ± 0.000000** | **89.37 ± 0.60** | **0.9749 ± 0.0005** | **0.8556 ± 0.0097** |
| 0.4 | OS | 0.477243 ± 0.000000 | 77.93 ± 1.00 | 0.9311 ± 0.0013 | 0.7453 ± 0.0145 |
| | RW | 0.450760 ± 0.000000 | 77.17 ± 0.68 | 0.9304 ± 0.0012 | 0.7374 ± 0.0119 |
| | SM | 0.477243 ± 0.000000 | 77.70 ± 1.01 | 0.9309 ± 0.0012 | 0.7409 ± 0.0137 |
| | ES | 0.450760 ± 0.000000 | 76.70 ± 0.78 | 0.9287 ± 0.0027 | 0.7322 ± 0.0105 |
| | GS | 0.450760 ± 0.000000 | 75.47 ± 2.75 | 0.9273 ± 0.0031 | 0.7176 ± 0.0379 |
| | GraphSHA | 0.716131 ± 0.000131 | 77.33 ± 0.00 | 0.9366 ± 0.0000 | 0.7302 ± 0.0001 |
| | LIMO | **0.988246 ± 0.000000** | **89.27 ± 0.55** | **0.9752 ± 0.0038** | **0.8530 ± 0.0113** |
| 0.2 | OS | 0.474473 ± 0.000000 | 73.83 ± 0.57 | 0.9227 ± 0.0007 | 0.6956 ± 0.0030 |
| | RW | 0.450760 ± 0.000000 | 73.80 ± 0.40 | 0.9214 ± 0.0020 | 0.6942 ± 0.0043 |
| | SM | 0.468297 ± 0.000000 | 71.67 ± 0.40 | 0.9158 ± 0.0002 | 0.6536 ± 0.0037 |
| | ES | 0.450760 ± 0.000000 | 73.73 ± 0.55 | 0.9203 ± 0.0044 | 0.6936 ± 0.0028 |
| | GS | 0.450760 ± 0.000000 | 71.67 ± 0.21 | 0.9139±0.0006 | 0.6552±0.0031 |
| | GraphSHA | 0.442478 ± 0.294456 | 73.00 ± 0.03 | 0.9320 ± 0.0000 | 0.7193 ± 0.0000 |
| | LIMO | **0.983798 ± 0.000000** | 87.60 ± 0.30 | **0.972044 ± 0.001620** | **0.8273 ± 0.0050** |
| 0.1 | OS | 0.468297 ± 0.000000 | 71.33 ± 0.31 | 0.9153 ± 0.0003 | 0.6536 ± 0.0043 |
| | RW | 0.450760 ± 0.000000 | 71.63 ± 0.46 | 0.9152 ± 0.0001 | 0.6535 ± 0.0051 |
| | SM | 0.468297 ± 0.000000 | 71.66 ± 0.40 | 0.9158 ± 0.0002 | 0.6536 ± 0.0037 |
| | ES | 0.450760 ± 0.000000 | 71.60 ± 0.40 | 0.9155 ± 0.0002 | 0.6544 ± 0.0049 |
| | GS | 0.450760 ± 0.000000 | 71.66 ± 0.21 | 0.9139 ± 0.0006 | 0.6552 ± 0.0031 |
| | GraphSHA | 0.490000 ± 0.360531 | 72.93 ± 0.36 | 0.9268 ± 0.0000 | 0.7098 ± 0.0020 |
| | LIMO | **0.978192 ± 0.000000** | **85.90 ± 0.36** | **0.9701 ± 0.0010** | **0.7963 ± 0.0057** |

Table 7: Table for the performance of baselines and LIMO on the CiteSeer Long Tail dataset with node classification using GCN

| Imbalance ratio | Setting | LI | ACC (%) | AUC-ROC | F1-score |
|---|---|---|---|---|---|
| 0.6 | OS | 0.446417 ± 0.000000 | 68.90 ± 0.44 | 0.890171 ± 0.002815 | 0.660476 ± 0.008331 |
|  | RW | 0.450760 ± 0.000000 | 69.30 ± 0.30 | 0.897712 ± 0.000724 | 0.669292 ± 0.003145 |
|  | SM | 0.478898 ± 0.000000 | 68.55 ± 0.49 | 0.889618 ± 0.006080 | 0.601887 ± 0.012084 |
|  | ES | 0.450760 ± 0.000000 | 70.17 ± 0.29 | 0.899685 ± 0.000626 | 0.657237 ± 0.005983 |
|  | GS | 0.450760 ± 0.000000 | 67.83 ± 0.42 | 0.890758 ± 0.006680 | 0.654973 ± 0.003506 |
|  | GraphSHA | 0.746710 ± 0.000000 | 77.93 ± 0.00 | 0.936423 ± 0.000024 | 0.748549 ± 0.000062 |
|  | **LIMO** | **0.990356 ± 0.000000** | **89.43 ± 0.12** | **0.976467 ± 0.000760** | **0.856456 ± 0.004074** |
| 0.5 | OS | 0.444640 ± 0.000000 | 68.30 ± 0.17 | 0.888627 ± 0.005262 | 0.653147 ± 0.006389 |
|  | RW | 0.450760 ± 0.000000 | 67.97 ± 0.25 | 0.894607 ± 0.002693 | 0.655509 ± 0.000167 |
|  | SM | 0.477518 ± 0.000000 | 66.70 ± 0.85 | 0.889663 ± 0.006083 | 0.588322 ± 0.015761 |
|  | ES | 0.450760 ± 0.000000 | 67.43 ± 0.15 | 0.895487 ± 0.001289 | 0.651248 ± 0.002490 |
|  | GS | 0.450526 ± 0.000330 | 65.35 ± 0.78 | 0.896441 ± 0.003702 | 0.631953 ± 0.001048 |
|  | GraphSHA | 0.753124 ± 0.000203 | 77.87 ± 0.00 | 0.938071 ± 0.000003 | 0.750878 ± 0.00002 |
|  | **LIMO** | **0.989453 ± 0.000000** | **87.87 ± 0.15** | **0.975472 ± 0.001425** | **0.841992 ± 0.001982** |
| 0.4 | OS | 0.443899 ± 0.000000 | 66.40 ± 0.26 | 0.886844 ± 0.002257 | 0.639946 ± 0.003119 |
|  | RW | 0.450760 ± 0.000000 | 66.33 ± 0.76 | 0.889157 ± 0.001822 | 0.639803 ± 0.006857 |
|  | SM | 0.477243 ± 0.000000 | 64.45 ± 0.07 | 0.888734 ± 0.005278 | 0.575310 ± 0.017942 |
|  | ES | 0.450760 ± 0.000000 | 66.23 ± 0.97 | 0.890453 ± 0.001315 | 0.639612 ± 0.008641 |
|  | GS | 0.449954 ± 0.000763 | 64.83 ± 1.50 | 0.893443 ± 0.007547 | 0.628159 ± 0.014430 |
|  | GraphSHA | 0.760120 ± 0.000235 | 77.23 ± 0.01 | 0.936290 ± 0.000003 | 0.743741 ± 0.000090 |
|  | **LIMO** | **0.988246 ± 0.000000** | **87.10 ± 0.52** | **0.976667 ± 0.000077** | **0.835901 ± 0.003445** |
| 0.2 | OS | 0.467733 ± 0.000000 | 59.97 ± 1.30 | 0.881278 ± 0.001615 | 0.527800 ± 0.011220 |
|  | RW | 0.450760 ± 0.000000 | 62.13 ± 1.64 | 0.888319 ± 0.007002 | 0.535145 ± 0.005992 |
|  | SM | 0.474473 ± 0.000000 | 59.00 ± 1.27 | 0.879820 ± 0.004675 | 0.531785 ± 0.013449 |
|  | ES | 0.450760 ± 0.000000 | 62.17 ± 1.82 | 0.887737 ± 0.007446 | 0.535795 ± 0.009713 |
|  | GS | 0.461211 ± 0.007053 | 60.13 ± 0.81 | 0.886106 ± 0.006330 | 0.554783 ± 0.020292 |
|  | GraphSHA | 0.744597 ± 0.000045 | 75.63 ± 0.00 | 0.929215 ± 0.000002 | 0.722743 ± 0.000029 |
|  | **LIMO** | **0.983798 ± 0.000000** | **83.37 ± 0.61** | **0.972302 ± 0.001058** | **0.785360 ± 0.010478** |
| 0.1 | OS | 0.463491 ± 0.000000 | 55.13 ± 0.15 | 0.865911 ± 0.009843 | 0.444481 ± 0.003753 |
|  | RW | 0.450760 ± 0.000000 | 57.20 ± 0.61 | 0.870522 ± 0.002306 | 0.462029 ± 0.003684 |
|  | SM | 0.468297 ± 0.000000 | 53.95 ± 0.21 | 0.857088 ± 0.004582 | 0.441853 ± 0.022925 |
|  | ES | 0.450760 ± 0.000000 | 57.57 ± 0.49 | 0.870388 ± 0.002294 | 0.464363 ± 0.003757 |
|  | GS | 0.460644 ± 0.011577 | 54.70 ± 0.42 | 0.872434 ± 0.000079 | 0.503474 ± 0.010128 |
|  | GraphSHA | 0.755621 ± 0.000022 | 73.57 ± 0.01 | 0.914589 ± 0.000014 | 0.700218 ± 0.000116 |
|  | **LIMO** | **0.978192 ± 0.000000** | **80.17 ± 0.75** | **0.965655 ± 0.002045** | **0.728264 ± 0.020225** |

Table 8: Table for the performance of baselines and LIMO on the PubMed dataset with node classification using GraphSAGE

| Imbalance ratio | Setting | LI | ACC (%) | AUC-ROC | F1-score |
|---|---|---|---|---|---|
| 0.6 | OS | 0.409324 ± 0.00011 | 72.73 ± 3.74 | 0.9088 ± 0.0289 | 0.7127 ± 0.0452 |
| | RW | 0.409284 ± 0.00000 | 76.16 ± 4.94 | 0.9123 ± 0.0236 | 0.7534 ± 0.0548 |
| | SM | 0.409284 ± 0.00000 | 69.09 ± 3.43 | 0.8841 ± 0.0226 | 0.6659 ± 0.0374 |
| | ES | 0.409415 ± 0.00001 | 69.70 ± 4.29 | 0.8867 ± 0.0191 | 0.6721 ± 0.0537 |
| | GS | 0.409284 ± 0.00000 | 70.00 ± 3.86 | 0.8898 ± 0.0165 | 0.6735 ± 0.0494 |
| | LIMO | **0.583905 ± 0.00000** | **92.73 ± 3.43** | **0.9804 ± 0.0115** | **0.9277 ± 0.0329** |
| 0.5 | OS | 0.409259 ± 0.00005 | 70.91 ± 4.45 | 0.9037 ± 0.0282 | 0.6851 ± 0.0610 |
| | RW | 0.409284 ± 0.00000 | 73.33 ± 5.55 | 0.8963 ± 0.0303 | 0.7140 ± 0.0712 |
| | SM | 0.409284 ± 0.00000 | 66.97 ± 2.14 | 0.8768 ± 0.0042 | 0.6347 ± 0.0328 |
| | ES | 0.409298 ± 0.00003 | 67.88 ± 1.71 | 0.8820 ± 0.0085 | 0.6441 ± 0.0204 |
| | GS | 0.409284 ± 0.00000 | 67.68 ± 2.45 | 0.8707 ± 0.0070 | 0.6421 ± 0.0345 |
| | LIMO | **0.57292 ± 0.00000** | **91.52 ± 3.43** | **0.9812 ± 0.0053** | **0.9155 ± 0.0345** |
| 0.4 | OS | 0.409301 ± 0.00003 | 71.52 ± 3.53 | 0.9046 ± 0.0222 | 0.6853 ± 0.0484 |
| | RW | 0.409284 ± 0.00000 | 72.12 ± 3.78 | 0.8992 ± 0.0336 | 0.6889 ± 0.0597 |
| | SM | 0.409284 ± 0.00000 | 68.18 ± 1.29 | 0.8763 ± 0.0242 | 0.6392 ± 0.0178 |
| | ES | 0.4093 ± 0.00003 | 67.27 ± 1.60 | 0.8628 ± 0.0215 | 0.6359 ± 0.0402 |
| | GS | 0.409284 ± 0.00000 | 67.07 ± 1.40 | 0.8798 ± 0.0060 | 0.6197 ± 0.0193 |
| | LIMO | **0.559164 ± 0.00000** | **91.52 ± 1.71** | **0.9814 ± 0.0032** | **0.9155 ± 0.0172** |
| 0.2 | OS | 0.409341 ± 0.00003 | 65.25 ± 2.13 | 0.8794 ± 0.0226 | 0.5750 ± 0.0439 |
| | RW | 0.409284 ± 0.00000 | 65.05 ± 4.13 | 0.8730 ± 0.0342 | 0.5724 ± 0.0738 |
| | SM | 0.409284 ± 0.00000 | 63.33 ± 0.43 | 0.8758 ± 0.0298 | 0.5433 ± 0.0188 |
| | ES | 0.409349 ± 0.00004 | 63.64 ± 0.86 | 0.8788 ± 0.0201 | 0.5418 ± 0.0286 |
| | GS | 0.409284 ± 0.00000 | 63.03 ± 0.00 | 0.8681 ± 0.0245 | 0.5344 ± 0.0091 |
| | LIMO | **0.516267 ± 0.00000** | **85.45 ± 0.86** | **0.9683 ± 0.0030** | **0.8528 ± 0.0087** |
| 0.1 | OS | 0.409322 ± 0.00002 | 63.03 ± 1.05 | 0.8684 ± 0.0301 | 0.5211 ± 0.0150 |
| | RW | 0.409284 ± 0.00000 | 64.85 ± 1.82 | 0.8801 ± 0.0118 | 0.5535 ± 0.0316 |
| | SM | 0.409284 ± 0.00000 | 62.42 ± 1.71 | 0.8603 ± 0.0085 | 0.5161 ± 0.0231 |
| | ES | 0.40933 ± 0.00003 | 62.73 ± 1.29 | 0.8508 ± 0.0125 | 0.5138 ± 0.0124 |
| | GS | 0.409284 ± 0.00000 | 62.83 ± 1.40 | 0.8587 ± 0.0032 | 0.5155 ± 0.0136 |
| | LIMO | **0.478199 ± 0.00000** | **68.18 ± 4.71** | **0.9279 ± 0.0068** | **0.6187 ± 0.0635** |

## C.2 Heterophilous Data

Here we have listed the results for the performance on some more heterophilous dataset, BlogCatalog, Twitter, Amazon (U-P-U), Amazon (U-S-U), Amazon (U-V-U), and Amazon (All). Here we observed in Amazon (ALL) and (U-S-U) that when the average of graph is already high LIMO could not perform as good as when the density is less, as in the other cases, for low imbalance ratios.

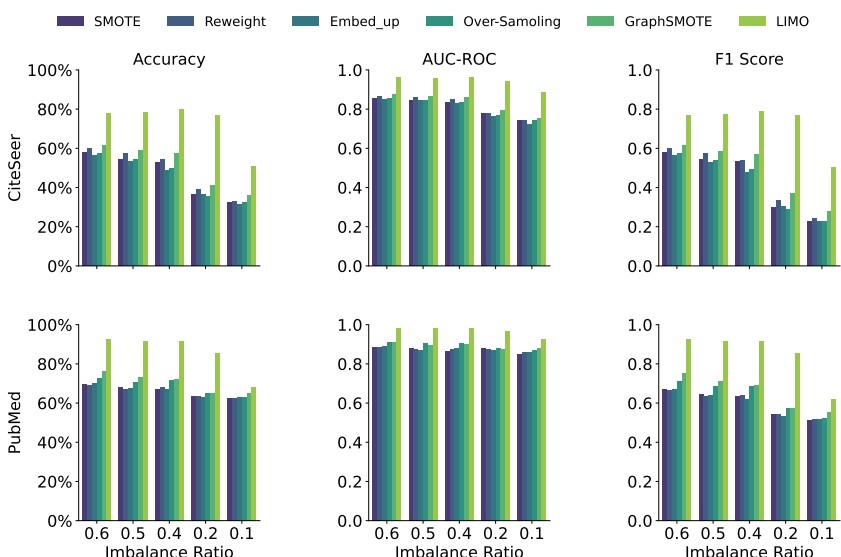

Figure 6: Performance of GNN on CiteSeer and PubMed datasets

Table 9: Table for the performance of baselines and LIMO on the BlogCatalog dataset with node classification using GraphSAGE

| Imbalance ratio | Setting | LI | ACC (%) | AUC-ROC | F1-score |
|---|---|---|---|---|---|
| 0.6 | OS | 0.010281 ± 0.000148 | 7.08 ± 0.58 | 0.5570 ± 0.0165 | 0.0646 ± 0.0046 |
| | RW | 0.010383 ± 0.000000 | 7.02 ± 0.74 | 0.5533 ± 0.0124 | 0.0629 ± 0.0069 |
| | SM | 0.010281 ± 0.000148 | 7.57 ± 0.12 | 0.5572 ± 0.0139 | 0.0692 ± 0.0024 |
| | ES | 0.010383 ± 0.000000 | 7.19 ± 0.24 | 0.5544 ± 0.0144 | 0.0656 ± 0.0023 |
| | GS | 0.010378 ± 0.000009 | 9.09 ± 0.56 | 0.5817 ± 0.0004 | 0.0821 ± 0.0069 |
| | LIMO | **0.098992 ± 0.000000** | **24.34 ± 1.01** | **0.8004 ± 0.0111** | **0.2473 ± 0.0088** |
| 0.5 | OS | 0.010306 ± 0.000115 | 7.25 ± 0.79 | 0.5541 ± 0.0098 | 0.0653 ± 0.0066 |
| | RW | 0.010383 ± 0.000000 | 7.30 ± 0.58 | 0.5506 ± 0.0111 | 0.0658 ± 0.0046 |
| | SM | 0.010306 ± 0.000115 | 7.64 ± 0.30 | 0.5558 ± 0.0108 | 0.0689 ± 0.0025 |
| | ES | 0.010383 ± 0.000000 | 7.81 ± 0.39 | 0.5529 ± 0.0122 | 0.0697 ± 0.0035 |
| | GS | 0.010370 ± 0.000009 | 8.94 ± 0.70 | 0.5707 ± 0.0110 | 0.0793 ± 0.0046 |
| | LIMO | **0.090304 ± 0.000000** | **21.49 ± 4.74** | **0.7571 ± 0.0679** | **0.2145 ± 0.0487** |
| 0.4 | OS | 0.010311 ± 0.000140 | 7.86 ± 0.61 | 0.5501 ± 0.0240 | 0.0700 ± 0.0071 |
| | RW | 0.010383 ± 0.000000 | 7.15 ± 0.38 | 0.5442 ± 0.0127 | 0.0641 ± 0.0049 |
| | SM | 0.010311 ± 0.000140 | 7.64 ± 0.30 | 0.5532 ± 0.0129 | 0.0730 ± 0.0018 |
| | ES | 0.010383 ± 0.000000 | 7.65 ± 0.20 | 0.5479 ± 0.0136 | 0.0674 ± 0.0023 |
| | GS | 0.010377 ± 0.000014 | 8.88 ± 1.02 | 0.5711 ± 0.0146 | 0.0806 ± 0.0095 |
| | LIMO | **0.079760 ± 0.000000** | **17.79 ± 6.65** | **0.7078 ± 0.0958** | **0.1709 ± 0.0684** |
| 0.2 | OS | 0.010340 ± 0.000033 | 7.78 ± 0.79 | 0.5437 ± 0.0084 | 0.0646 ± 0.0078 |
| | RW | 0.010383 ± 0.000000 | 7.84 ± 0.21 | 0.5450 ± 0.0064 | 0.0649 ± 0.0025 |
| | SM | 0.010340 ± 0.000033 | 7.63 ± 0.30 | 0.5454 ± 0.0083 | 0.0646 ± 0.0043 |
| | ES | 0.010383 ± 0.000000 | 8.02 ± 0.20 | 0.5438 ± 0.0058 | 0.0661 ± 0.0027 |
| | GS | 0.010381 ± 0.000003 | 9.01 ± 0.82 | 0.5721 ± 0.0088 | 0.0761 ± 0.0101 |
| | LIMO | **0.050332 ± 0.000000** | **13.17 ± 2.11** | **0.6563 ± 0.0522** | **0.1114 ± 0.0225** |
| 0.1 | OS | 0.010370 ± 0.000034 | 7.49 ± 1.08 | 0.5398 ± 0.0093 | 0.0605 ± 0.0106 |
| | RW | 0.010383 ± 0.000000 | 7.51 ± 0.51 | 0.5415 ± 0.0055 | 0.0602 ± 0.0059 |
| | SM | 0.010370 ± 0.000034 | 7.51 ± 0.33 | 0.5431 ± 0.0068 | 0.0605 ± 0.0018 |
| | ES | 0.010383 ± 0.000000 | 7.62 ± 0.36 | 0.5404 ± 0.0028 | 0.0608 ± 0.0038 |
| | GS | 0.010380 ± 0.000005 | 9.04 ± 0.61 | 0.5703 ± 0.0009 | 0.0721 ± 0.0046 |
| | LIMO | **0.029974 ± 0.000000** | **12.48 ± 1.44** | **0.6281 ± 0.0302** | **0.0977 ± 0.0137** |

Table 10: Table for the performance of baselines and LIMO on the Twitter dataset with node classification using GraphSAGE

| Imbalance ratio | Setting | LI | ACC (%) | AUC-ROC | F1-score |
|---|---|---|---|---|---|
| 0.6 | OS | 0.000015 ± 0.000000 | 48.79 ± 4.10 | 0.5259 ± 0.0311 | 0.4864 ± 0.0399 |
| | RW | 0.000012 ± 0.000000 | 51.21 ± 1.39 | 0.5158 ± 0.0358 | 0.5088 ± 0.0144 |
| | SM | 0.000015 ± 0.000000 | 49.39 ± 1.89 | 0.5086 ± 0.0221 | 0.4923 ± 0.0213 |
| | ES | 0.000012 ± 0.000000 | 51.21 ± 1.05 | 0.5061 ± 0.0237 | 0.5118 ± 0.0102 |
| | GS | 0.000012 ± 0.000002 | 53.33 ± 5.48 | 0.5466 ± 0.0889 | 0.5273 ± 0.0617 |
| | LIMO | **0.083961 ± 0.000000** | **68.79 ± 5.01** | **0.7236 ± 0.0827** | **0.6849 ± 0.0529** |
| 0.5 | OS | 0.000015 ± 0.000003 | 49.70 ± 2.10 | 0.5052 ± 0.0367 | 0.4955 ± 0.0211 |
| | RW | 0.000012 ± 0.000000 | 51.21 ± 1.39 | 0.5262 ± 0.0578 | 0.5089 ± 0.0142 |
| | SM | 0.000014 ± 0.000002 | 50.91 ± 0.91 | 0.5046 ± 0.0018 | 0.5043 ± 0.0110 |
| | ES | 0.000012 ± 0.000000 | 49.09 ± 0.91 | 0.4988 ± 0.0182 | 0.4729 ± 0.0268 |
| | GS | 0.000013 ± 0.000001 | 51.51 ± 3.67 | 0.5098 ± 0.0293 | 0.4740 ± 0.0747 |
| | LIMO | **0.069097 ± 0.000000** | **67.27 ± 5.06** | **0.7031 ± 0.0692** | **0.6668 ± 0.0510** |
| 0.4 | OS | 0.000016 ± 0.000003 | 49.70 ± 1.39 | 0.4969 ± 0.0244 | 0.4931 ± 0.0101 |
| | RW | 0.000012 ± 0.000000 | 51.21 ± 2.78 | 0.5089 ± 0.0179 | 0.5042 ± 0.0208 |
| | SM | 0.000016 ± 0.000003 | 52.12 ± 1.39 | 0.5141 ± 0.0077 | 0.5120 ± 0.0141 |
| | ES | 0.000012 ± 0.000000 | 53.03 ± 1.05 | 0.5288 ± 0.0499 | 0.5122 ± 0.0094 |
| | GS | 0.000013 ± 0.000001 | 57.57 ± 2.29 | 0.5779 ± 0.0416 | 0.5697 ± 0.0161 |
| | LIMO | **0.053989 ± 0.000000** | **64.24 ± 2.77** | **0.6258 ± 0.0413** | **0.6368 ± 0.0299** |
| 0.2 | OS | 0.000014 ± 0.000000 | 48.49 ± 5.33 | 0.4847 ± 0.0196 | 0.4415 ± 0.0434 |
| | RW | 0.000012 ± 0.000000 | 52.73 ± 5.06 | 0.4895 ± 0.0593 | 0.4729 ± 0.0495 |
| | SM | 0.000014 ± 0.000000 | 53.34 ± 2.10 | 0.4977 ± 0.0609 | 0.4550 ± 0.0560 |
| | ES | 0.000012 ± 0.000000 | 53.16 ± 3.54 | 0.5036 ± 0.0832 | 0.4389 ± 0.0699 |
| | GS | 0.000012 ± 0.000000 | 52.42 ± 4.29 | 0.5342 ± 0.0630 | 0.4488 ± 0.0359 |
| | LIMO | **0.023656 ± 0.000000** | **57.88 ± 5.33** | **0.5993 ± 0.0646** | **0.5329 ± 0.0617** |
| 0.1 | OS | 0.000013 ± 0.000001 | 48.79 ± 2.93 | 0.4583 ± 0.0305 | 0.4114 ± 0.0279 |
| | RW | 0.000012 ± 0.000000 | 50.00 ± 3.86 | 0.4772 ± 0.0278 | 0.4367 ± 0.0136 |
| | SM | 0.000013 ± 0.000001 | 49.39 ± 1.89 | 0.4575 ± 0.0343 | 0.4010 ± 0.0421 |
| | ES | 0.000012 ± 0.000000 | 50.00 ± 2.41 | 0.4642 ± 0.0283 | 0.4042 ± 0.0380 |
| | GS | 0.000012 ± 0.000000 | 49.70 ± 0.53 | 0.4713 ± 0.0171 | 0.3790 ± 0.0404 |
| | LIMO | **0.009405 ± 0.000000** | **54.24 ± 1.39** | **0.4901 ± 0.0179** | **0.4729 ± 0.0317** |

Table 11: Table for the performance of baselines and LIMO on the Amazon (U-P-U) dataset with node classification using GraphSAGE

| Imbalance ratio | Setting | LI | ACC (%) | AUC-ROC | F1-score |
|---|---|---|---|---|---|
| 0.6 | OS | 0.004346 ± 0.00005 | 80.30 ± 4.20 | 0.8882 ± 0.0244 | 0.8025 ± 0.0424 |
| | RW | 0.004334 ± 0.00000 | 77.73 ± 5.79 | 0.8868 ± 0.0456 | 0.7764 ± 0.0591 |
| | SM | 0.004322 ± 0.00001 | 66.55 ± 4.47 | 0.8120 ± 0.0430 | 0.6539 ± 0.0488 |
| | ES | 0.004334 ± 0.00000 | 77.73 ± 5.79 | 0.8883 ± 0.0538 | 0.7763 ± 0.0589 |
| | GS | 0.004363 ± 0.00005 | 80.91 ± 4.55 | 0.9006 ± 0.0290 | 0.8082 ± 0.0454 |
| | **LIMO** | **0.121393 ± 0.00000** | **92.12 ± 1.05** | **0.9835 ± 0.0006** | **0.9210 ± 0.0107** |
| 0.5 | OS | 0.004345 ± 0.00005 | 83.64 ± 3.28 | 0.9068 ± 0.0233 | 0.8361 ± 0.0329 |
| | RW | 0.004334 ± 0.00000 | 84.55 ± 0.00 | 0.9124 ± 0.0000 | 0.8454 ± 0.0000 |
| | SM | 0.004317 ± 0.00001 | 83.18 ± 1.93 | 0.9136 ± 0.0068 | 0.8307 ± 0.0208 |
| | ES | 0.004334 ± 0.00000 | 80.00 ± 3.86 | 0.9013 ± 0.0222 | 0.7992 ± 0.0396 |
| | GS | 0.004366 ± 0.00006 | 80.30 ± 2.78 | 0.8988 ± 0.0285 | 0.8027 ± 0.0279 |
| | **LIMO** | **0.097429 ± 0.00000** | **92.12 ± 1.05** | **0.9727 ± 0.0122** | **0.9210 ± 0.0104** |
| 0.4 | OS | 0.004354 ± 0.00005 | 82.73 ± 4.17 | 0.9060 ± 0.0114 | 0.8265 ± 0.0420 |
| | RW | 0.004334 ± 0.00000 | 83.18 ± 0.64 | 0.9073 ± 0.0157 | 0.8315 ± 0.0065 |
| | SM | 0.004327 ± 0.00000 | 83.18 ± 5.79 | 0.9136 ± 0.0091 | 0.8309 ± 0.0586 |
| | ES | 0.004334 ± 0.00000 | 81.82 ± 3.86 | 0.9116 ± 0.0110 | 0.8178 ± 0.0386 |
| | GS | 0.004334 ± 0.00000 | 84.24 ± 3.67 | 0.9142 ± 0.0209 | 0.8417 ± 0.0362 |
| | **LIMO** | **0.072666 ± 0.00000** | **91.82 ± 2.57** | **0.9765 ± 0.0112** | **0.9181 ± 0.0257** |
| 0.2 | OS | 0.00434 ± 0.00002 | 78.48 ± 5.48 | 0.8454 ± 0.0219 | 0.7804 ± 0.0560 |
| | RW | 0.004334 ± 0.00000 | 74.55 ± 0.00 | 0.8519 ± 0.0000 | 0.7400 ± 0.0000 |
| | SM | 0.004339 ± 0.00002 | 75.45 ± 5.14 | 0.8339 ± 0.0157 | 0.7491 ± 0.0535 |
| | ES | 0.004334 ± 0.00000 | 80.00 ± 3.86 | 0.8493 ± 0.0262 | 0.7972 ± 0.0379 |
| | GS | 0.004334 ± 0.00000 | 80.30 ± 4.20 | 0.8716 ± 0.0344 | 0.7995 ± 0.0442 |
| | **LIMO** | **0.023651 ± 0.00000** | **85.15 ± 2.62** | **0.9408 ± 0.0103** | **0.8496 ± 0.0264** |
| 0.1 | OS | 0.004337 ± 0.00002 | **70.61 ± 18.42** | 0.7750 ± 0.1862 | **0.6598 ± 0.2506** |
| | RW | 0.004334 ± 0.00000 | 50.00 ± 0.00 | 0.5666 ± 0.0000 | 0.3762 ± 0.0000 |
| | SM | 0.004342 ± 0.00002 | 65.45 ± 21.86 | 0.7013 ± 0.1947 | 0.5908 ± 0.3035 |
| | ES | 0.004334 ± 0.00000 | 65.91 ± 22.50 | 0.7018 ± 0.1907 | 0.5952 ± 0.3097 |
| | GS | 0.004334 ± 0.00000 | 69.39 ± 16.89 | 0.6993 ± 0.3016 | 0.6325 ± 0.2599 |
| | **LIMO** | **0.004509 ± 0.00000** | 65.45 ± 16.69 | **0.8188 ± 0.0757** | 0.6016 ± 0.2099 |

Table 12: Table for the performance of baselines and LIMO on the Amazon (U-S-U) dataset with node classification using GraphSAGE

| Imbalance ratio | Setting | LI | ACC (%) | AUC-ROC | F1-score |
|---|---|---|---|---|---|
| 0.6 | OS | 0.00379 ± 0.00016 | 83.33 ± 1.05 | 0.9116 ± 0.0329 | 0.8331 ± 0.0106 |
| | RW | 0.00387 ± 0.00000 | 84.85 ± 2.10 | 0.9163 ± 0.0364 | 0.8481 ± 0.0212 |
| | SM | 0.00379 ± 0.00016 | 83.94 ± 2.78 | 0.9148 ± 0.0357 | 0.8390 ± 0.0275 |
| | ES | 0.00387 ± 0.00000 | 85.15 ± 1.89 | 0.9172 ± 0.0381 | 0.8511 ± 0.0191 |
| | GS | 0.003954 ± 0.00007 | 81.21 ± 5.01 | 0.9230 ± 0.0307 | 0.8098 ± 0.0533 |
| | LIMO | **0.038549 ± 0.00000** | **88.79 ± 1.89** | **0.9468 ± 0.0476** | **0.8874 ± 0.0191** |
| 0.5 | OS | 0.003794 ± 0.00014 | 84.85 ± 3.67 | 0.9148 ± 0.0319 | 0.8480 ± 0.0372 |
| | RW | 0.00387 ± 0.00000 | 85.15 ± 2.78 | 0.9129 ± 0.0343 | 0.8509 ± 0.0283 |
| | SM | 0.003794 ± 0.00014 | 86.06 ± 1.89 | 0.9161 ± 0.0363 | 0.8602 ± 0.0192 |
| | ES | 0.00387 ± 0.00000 | 85.45 ± 2.41 | 0.9126 ± 0.0337 | 0.8540 ± 0.0245 |
| | GS | 0.003959 ± 0.00007 | 84.24 ± 1.05 | 0.9188 ± 0.0279 | 0.8419 ± 0.0108 |
| | LIMO | **0.031885 ± 0.00000** | **90.00 ± 1.82** | **0.9431 ± 0.0490** | **0.8996 ± 0.0186** |
| 0.4 | OS | 0.003845 ± 0.00013 | 85.76 ± 2.29 | 0.9113 ± 0.0406 | 0.8571 ± 0.0230 |
| | RW | 0.00387 ± 0.00000 | 86.36 ± 2.41 | 0.9146 ± 0.0417 | 0.8632 ± 0.0244 |
| | SM | 0.003845 ± 0.00013 | 86.97 ± 1.89 | **0.9229 ± 0.0347** | 0.8694 ± 0.0188 |
| | ES | 0.00387 ± 0.00000 | 85.45 ± 1.82 | 0.9107 ± 0.0407 | 0.8541 ± 0.0184 |
| | GS | 0.003912 ± 0.00007 | 85.45 ± 0.91 | 0.9211 ± 0.0301 | 0.8542 ± 0.0094 |
| | LIMO | **0.025419 ± 0.00000** | **88.79 ± 1.05** | 0.9215 ± 0.0159 | **0.8873 ± 0.0106** |
| 0.2 | OS | 0.003836 ± 0.00008 | 81.52 ± 5.33 | 0.8552 ± 0.0202 | 0.8098 ± 0.0602 |
| | RW | 0.00387 ± 0.00000 | 81.82 ± 4.81 | 0.8565 ± 0.0185 | 0.8133 ± 0.0541 |
| | SM | 0.003836 ± 0.00008 | 81.21 ± 5.84 | 0.8519 ± 0.0127 | 0.8064 ± 0.0665 |
| | ES | 0.00387 ± 0.00000 | 82.12 ± 5.17 | 0.8558 ± 0.0159 | 0.8165 ± 0.0577 |
| | GS | 0.003895 ± 0.00004 | 81.52 ± 7.06 | 0.8517 ± 0.0496 | 0.8120 ± 0.0734 |
| | LIMO | **0.013454 ± 0.00000** | **82.73 ± 4.55** | **0.8720 ± 0.0269** | **0.8241 ± 0.0475** |
| 0.1 | OS | 0.003826 ± 0.00008 | 74.85 ± 24.82 | 0.7251 ± 0.3064 | **0.7034 ± 0.3238** |
| | RW | 0.00387 ± 0.00000 | 74.85 ± 24.82 | 0.7535 ± 0.2575 | **0.7034 ± 0.3238** |
| | SM | 0.003826 ± 0.00008 | 74.85 ± 24.82 | 0.7418 ± 0.3231 | **0.7034 ± 0.3238** |
| | ES | 0.00387 ± 0.00000 | 74.85 ± 24.82 | 0.7280 ± 0.3015 | **0.7034 ± 0.3238** |
| | GS | 0.003871 ± 0.00000 | 73.03 ± 20.03 | **0.8130 ± 0.1763** | 0.6721 ± 0.2940 |
| | LIMO | **0.008241 ± 0.00000** | **75.45 ± 22.34** | 0.8055 ± 0.1675 | 0.7030 ± 0.3090 |

Table 13: Table for the performance of baselines and LIMO on the Amazon (U-V-U) dataset with node classification using GraphSAGE

| Imbalance ratio | Setting | LI | ACC (%) | AUC-ROC | F1-score |
|---|---|---|---|---|---|
| 0.6 | OS | 0.00514 ± 0.00004 | 86.36 ± 5.06 | 0.9404 ± 0.0166 | 0.8620 ± 0.0524 |
| | RW | 0.005175 ± 0.00000 | 84.24 ± 5.25 | 0.9415 ± 0.0259 | 0.8401 ± 0.0557 |
| | SM | 0.00514 ± 0.00004 | 86.67 ± 4.30 | 0.9468 ± 0.0160 | 0.8652 ± 0.0451 |
| | ES | 0.005175 ± 0.00000 | 84.85 ± 6.39 | 0.9209 ± 0.0455 | 0.8460 ± 0.0669 |
| | GS | 0.005269 ± 0.00005 | 85.76 ± 6.94 | 0.9382 ± 0.0172 | 0.8548 ± 0.0734 |
| | **LIMO** | **0.188869 ± 0.00000** | **91.82 ± 3.28** | **0.9835 ± 0.0056** | **0.9181 ± 0.0328** |
| 0.5 | OS | 0.005176 ± 0.00007 | 84.85 ± 4.67 | 0.9269 ± 0.0442 | 0.8474 ± 0.0479 |
| | RW | 0.005175 ± 0.00000 | 86.36 ± 5.06 | 0.9355 ± 0.0166 | 0.8621 ± 0.0524 |
| | SM | 0.005176 ± 0.00007 | 85.76 ± 4.67 | 0.9288 ± 0.0457 | 0.8566 ± 0.0476 |
| | ES | 0.005175 ± 0.00000 | 85.76 ± 4.67 | 0.9194 ± 0.0442 | 0.8560 ± 0.0480 |
| | GS | 0.00524 ± 0.00006 | 86.06 ± 6.19 | 0.9326 ± 0.0101 | 0.8585 ± 0.0645 |
| | **LIMO** | **0.161145 ± 0.00000** | **91.21 ± 2.92** | **0.9780 ± 0.0076** | **0.9120 ± 0.0292** |
| 0.4 | OS | 0.005199 ± 0.00009 | 86.97 ± 1.39 | 0.9203 ± 0.0243 | 0.8689 ± 0.0139 |
| | RW | 0.005175 ± 0.00000 | 86.36 ± 3.15 | 0.9319 ± 0.0047 | 0.8626 ± 0.0321 |
| | SM | 0.005199 ± 0.00009 | 86.06 ± 3.67 | 0.9307 ± 0.0047 | 0.8602 ± 0.0364 |
| | ES | 0.005175 ± 0.00000 | 85.45 ± 3.64 | 0.9214 ± 0.0350 | 0.8536 ± 0.0371 |
| | GS | 0.005248 ± 0.00006 | 85.45 ± 5.53 | 0.9254 ± 0.0104 | 0.8534 ± 0.0561 |
| | **LIMO** | **0.131436 ± 0.00000** | **90.91 ± 2.73** | **0.9601 ± 0.0205** | **0.9089 ± 0.0273** |
| 0.2 | OS | 0.005153 ± 0.00008 | 83.64 ± 2.41 | 0.8744 ± 0.0570 | 0.8329 ± 0.0265 |
| | RW | 0.005175 ± 0.00000 | 86.36 ± 3.15 | 0.8832 ± 0.0594 | 0.8618 ± 0.0328 |
| | SM | 0.005175 ± 0.00008 | 85.68 ± 1.36 | 0.8698 ± 0.0471 | 0.8551 ± 0.0141 |
| | ES | 0.005175 ± 0.00000 | 86.06 ± 3.78 | 0.8960 ± 0.0669 | 0.8586 ± 0.0394 |
| | GS | 0.005178 ± 0.00009 | 84.55 ± 3.28 | 0.8908 ± 0.0583 | 0.8436 ± 0.0341 |
| | **LIMO** | **0.066697 ± 0.00000** | **87.27 ± 4.17** | **0.9194 ± 0.0715** | **0.8713 ± 0.0432** |
| 0.1 | OS | 0.00516 ± 0.00008 | 73.03 ± 19.16 | 0.8337 ± 0.1311 | 0.6788 ± 0.2715 |
| | RW | 0.005175 ± 0.00000 | 73.64 ± 19.94 | 0.8360 ± 0.1315 | 0.6851 ± 0.2788 |
| | SM | 0.00516 ± 0.00008 | 73.33 ± 19.55 | 0.8365 ± 0.1359 | 0.6820 ± 0.2751 |
| | ES | 0.005175 ± 0.00000 | 73.33 ± 19.84 | 0.8331 ± 0.1293 | 0.6817 ± 0.2773 |
| | GS | 0.005193 ± 0.00003 | 73.03 ± 20.27 | 0.8182 ± 0.1871 | 0.6722 ± 0.2960 |
| | **LIMO** | **0.033485 ± 0.00000** | **74.55 ± 20.31** | **0.8451 ± 0.1245** | **0.6948 ± 0.2840** |

Table 14: Table for the performance of baselines and LIMO on the Amazon (All) dataset with node classification using GraphSAGE

| Imbalance ratio | Setting | LI | ACC (%) | AUC-ROC | F1-score |
|---|---|---|---|---|---|
| 0.6 | OS | 0.006686 ± 0.00024 | 82.42 ± 4.10 | 0.9402 ± 0.0347 | 0.8229 ± 0.0407 |
| | RW | 0.006798 ± 0.00000 | 88.64 ± 4.50 | 0.9412 ± 0.0112 | 0.8861 ± 0.0449 |
| | SM | 0.006548 ± 0.00007 | 85.00 ± 0.64 | 0.9602 ± 0.0147 | 0.8493 ± 0.0069 |
| | ES | 0.006798 ± 0.00000 | 87.27 ± 3.15 | 0.9256 ± 0.0425 | 0.8721 ± 0.0316 |
| | GS | 0.006949 ± 0.00016 | 87.58 ± 2.10 | 0.9369 ± 0.0349 | 0.8754 ± 0.0211 |
| | **LIMO** | **0.03222 ± 0.00000** | **92.73 ± 3.86** | **0.9736 ± 0.0061** | **0.9271 ± 0.0389** |
| 0.5 | OS | 0.0067 ± 0.00021 | 84.55 ± 2.73 | 0.9388 ± 0.0326 | 0.8447 ± 0.0275 |
| | RW | 0.006798 ± 0.00000 | 85.91 ± 1.93 | 0.9552 ± 0.0068 | 0.8583 ± 0.0204 |
| | SM | 0.006582 ± 0.00008 | 87.73 ± 4.50 | 0.9383 ± 0.0395 | 0.8765 ± 0.0454 |
| | ES | 0.006798 ± 0.00000 | 86.06 ± 1.39 | 0.9252 ± 0.0436 | 0.8596 ± 0.0146 |
| | GS | 0.006887 ± 0.00010 | 85.15 ± 0.52 | 0.9437 ± 0.0379 | 0.8503 ± 0.0061 |
| | **LIMO** | **0.027479 ± 0.00000** | **94.09 ± 0.64** | **0.9774 ± 0.0026** | **0.9409 ± 0.0064** |
| 0.4 | OS | 0.006771 ± 0.00021 | 85.45 ± 3.15 | 0.9193 ± 0.0224 | 0.8539 ± 0.0317 |
| | RW | 0.006798 ± 0.00000 | 88.64 ± 1.93 | 0.9202 ± 0.0475 | 0.8857 ± 0.0199 |
| | SM | 0.006685 ± 0.00020 | 85.00 ± 4.50 | 0.9312 ± 0.0005 | 0.8495 ± 0.0456 |
| | ES | 0.006798 ± 0.00000 | 85.76 ± 1.89 | 0.9158 ± 0.0259 | 0.8569 ± 0.0190 |
| | GS | 0.006865 ± 0.00007 | 85.76 ± 1.39 | 0.9320 ± 0.0181 | 0.8572 ± 0.0142 |
| | **LIMO** | **0.022876 ± 0.00000** | **90.45 ± 3.21** | **0.9544 ± 0.0103** | **0.9043 ± 0.0325** |
| 0.2 | OS | 0.006742 ± 0.00011 | **84.55 ± 5.06** | 0.8793 ± 0.0558 | 0.8432 ± 0.0521 |
| | RW | 0.006798 ± 0.00000 | 82.73 ± 3.86 | 0.8661 ± 0.0673 | 0.8250 ± 0.0415 |
| | SM | 0.006704 ± 0.00012 | 84.09 ± 4.50 | 0.8579 ± 0.0276 | 0.8390 ± 0.0462 |
| | ES | 0.006798 ± 0.00000 | 83.94 ± 5.55 | 0.8680 ± 0.0438 | 0.8372 ± 0.0569 |
| | GS | 0.006822 ± 0.00002 | **84.55 ± 0.91** | **0.8830 ± 0.0619** | **0.8439 ± 0.0107** |
| | **LIMO** | **0.014247 ± 0.00000** | 75.91 ± 7.07 | 0.8529 ± 0.0795 | 0.7489 ± 0.0768 |
| 0.1 | OS | 0.006735 ± 0.00011 | 72.73 ± 19.73 | 0.6962 ± 0.2811 | 0.6698 ± 0.2918 |
| | RW | 0.006798 ± 0.00000 | **85.45 ± 2.57** | **0.8762 ± 0.0563** | **0.8515 ± 0.0278** |
| | SM | 0.006702 ± 0.00013 | 67.73 ± 25.07 | 0.5990 ± 0.3170 | 0.5927 ± 0.3669 |
| | ES | 0.006798 ± 0.00000 | 74.24 ± 20.99 | 0.7386 ± 0.2774 | 0.6854 ± 0.3049 |
| | GS | 0.006802 ± 0.00001 | 72.12 ± 19.16 | 0.7520 ± 0.2205 | 0.6639 ± 0.2863 |
| | **LIMO** | **0.010334 ± 0.00000** | 66.82 ± 23.78 | 0.7478 ± 0.1099 | 0.5831 ± 0.3532 |

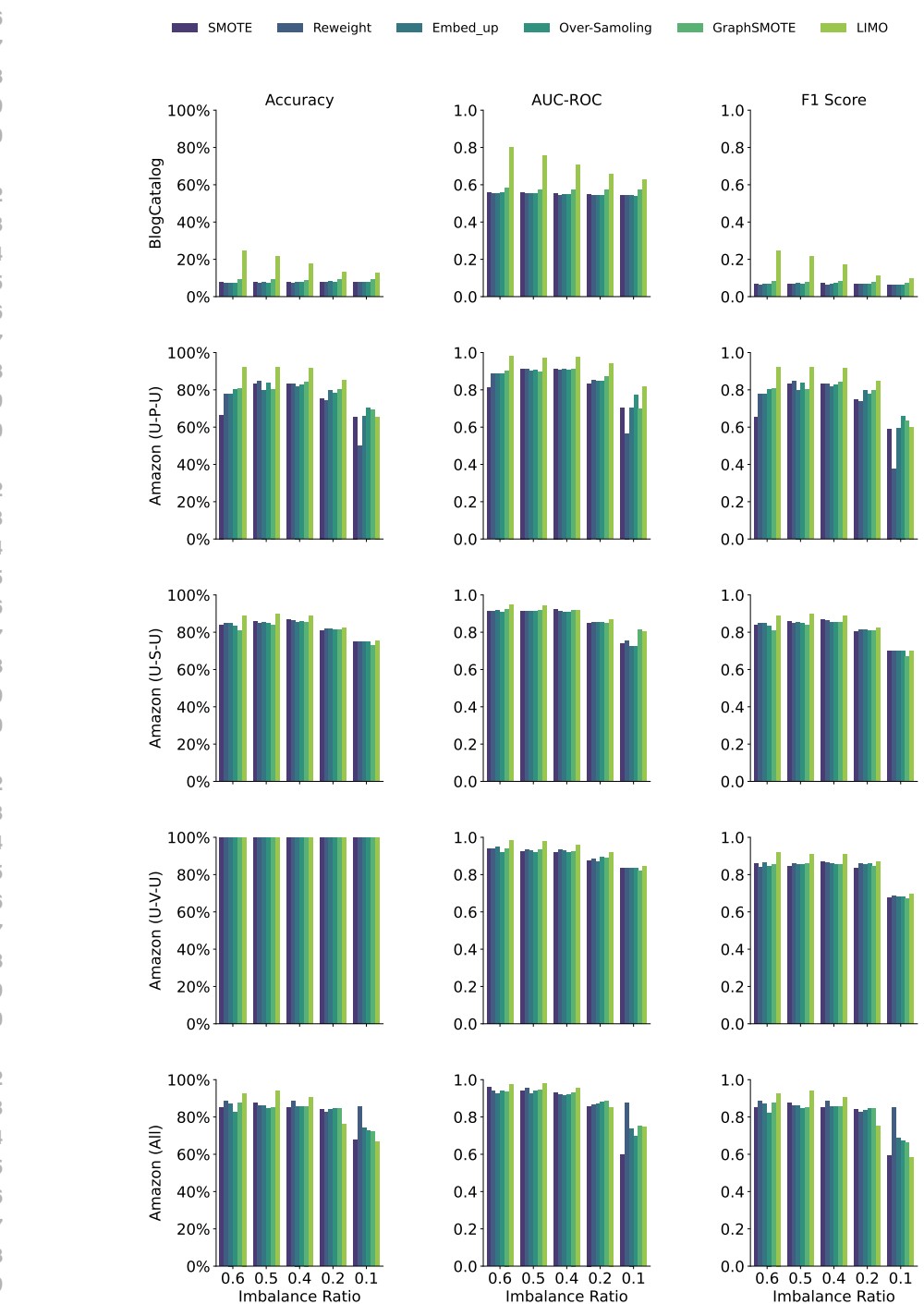

Figure 7: Performance on Heterophilous Datasets

## C.3 PARAMETER SENITICITY

To support the inference in section 5.4 we have provide results of the same experiment on the Cite-Seer dataset.

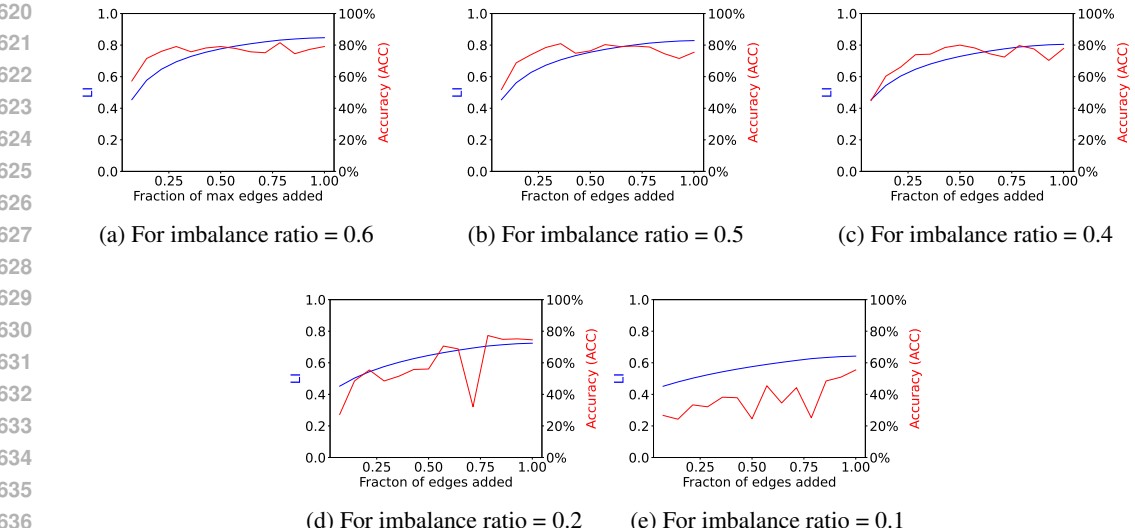

(a) For imbalance ratio = 0.6    (b) For imbalance ratio = 0.5    (c) For imbalance ratio = 0.4

(d) For imbalance ratio = 0.2    (e) For imbalance ratio = 0.1

Figure 8: The plots for the performance of GNN on CitesSeer dataset versus the node upscale factor

### C.4 ABLATION

This section contains the results for abalation study on CiteSeer dataset to support the fact that the LI is directly prorportional to performance of GNNs.

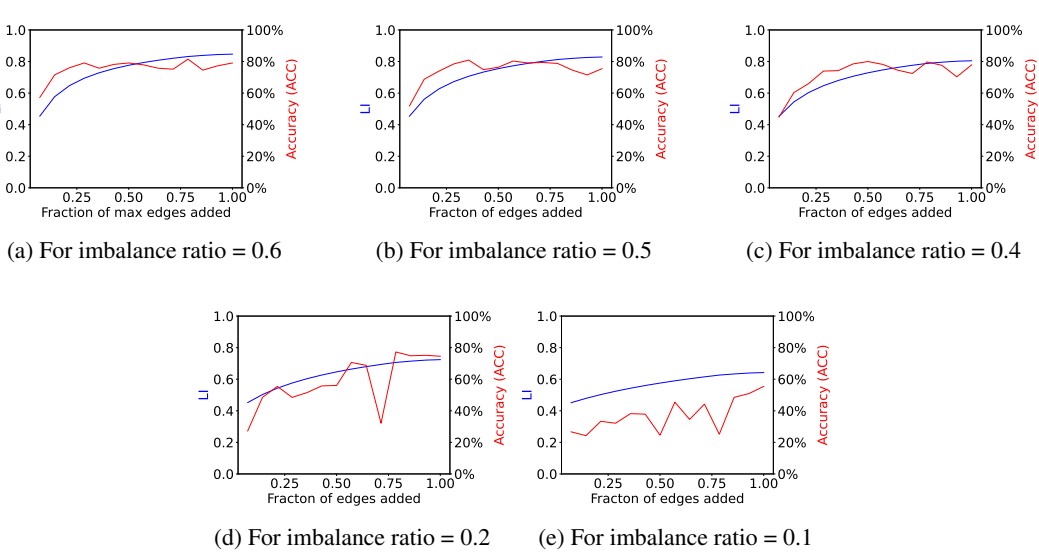

(a) For imbalance ratio = 0.6    (b) For imbalance ratio = 0.5    (c) For imbalance ratio = 0.4

(d) For imbalance ratio = 0.2    (e) For imbalance ratio = 0.1

Figure 9: The plots for the performance of GNN on CitesSeer dataset with an increase in LI

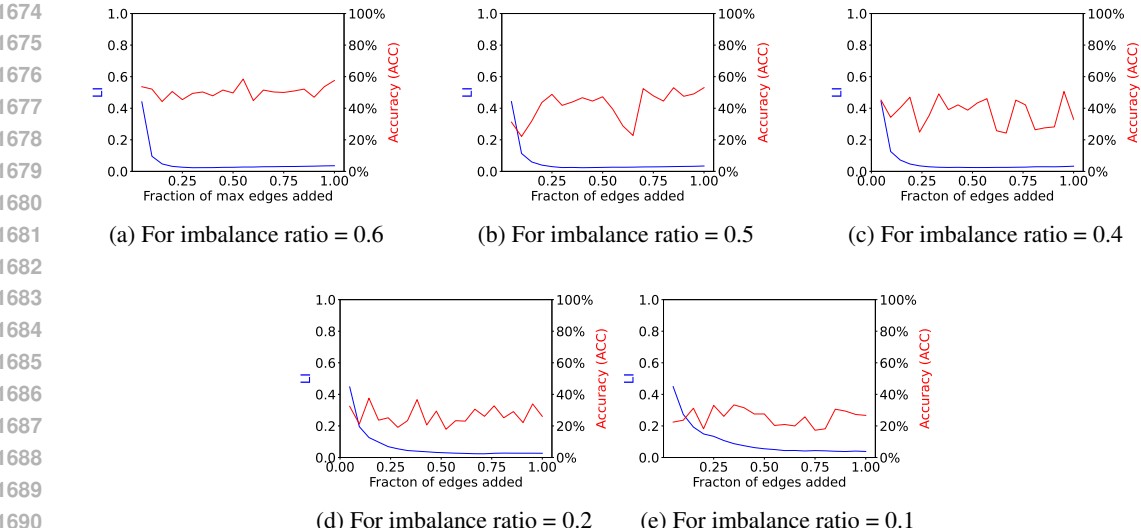

(a) For imbalance ratio = 0.6    (b) For imbalance ratio = 0.5    (c) For imbalance ratio = 0.4

(d) For imbalance ratio = 0.2    (e) For imbalance ratio = 0.1

Figure 10: The plots for the performance of GNN on CiteSeer dataset with a decrease in LI

