# OpenReview forum: "Label Informativeness-based Minority Oversampling in Graphs (LIMO)"
_ICLR.cc/2025/Conference — Submitted to ICLR 2025_

### Official Review · Reviewer_C2ac · 2024-10-28

**Soundness:** 2
**Presentation:** 2
**Contribution:** 2
**Rating:** 5
**Confidence:** 4

**Summary:**

This paper investigates class imbalanced graph representation learning, where certain classes have significantly fewer nodes. Ignoring such kind of information will result in biased learning or overfitting. The authors claim that existing oversampling techniques overlook the label informativeness of the graphs, which measures the amount of information regarding to its neighbor’s labels. To this end, the authors propose LIMO to oversample minority class nodes by maximizing LI. Synthetic node features are generated with SMOTE, then edges are added by analyzing the derivative of its LI. Comprehensive experiments on both homophilous and heterophilous datasets demonstrate that the proposed model can outperform recent baselines under different imbalance ratios.

**Strengths:**

1.	The paper is generally well-written and easy to understand, meanwhile, the results appear to be promising.

2.	Experiments are conducted on both homophilous and heterophilous datasets, which broadens its application.

3.	Various ablation studies are given to show the impacts of the synthesized nodes and generated edges.

**Weaknesses:**

1.	The calculation of $t$ value given in Theorem 1 is not rigorous, since the authors explore it in a certain range using predefined steps, which might not be the optimal value. The proof of Theorem 2 is also not convincing.

2.	The chosen baselines are too old, and some key baselines are not discussed and compared, e.g., GraphENS, and GraphSHA, etc.

3.	No time and space complexity analyses are given to show the efficiency of the proposed model.

**Questions:**

1.	The calculation of $t$ value given in Theorem 1 is not rigorous, since the authors explore it in a certain range using predefined steps, which might not be the optimal value. The proof of Theorem 2 is also not convincing.

2.	The chosen baselines are too old, and some key baselines are not discussed and compared, e.g., GraphENS [1], and GraphSHA [2], etc.

3.	No time and space complexity analyses are given to show the efficiency of the proposed model.

4.	The authors claim that the proposed model does not significantly increase the data volume. However, the authors synthesize nodes and edges in the graph, which indeed increases the data volume.

5.	The caption in Figure 1 does not provide any information, and the authors did not cite references appropriately. For instance, SMOTE was not cited upon its first mention.

6.	What labels are used in the Amazon datasets? The original datasets do not include labels for the user nodes.


References:

[1] Park J, Song J, Yang E. Graphens: Neighbor-aware ego network synthesis for class-imbalanced node classification[C]//International conference on learning representations. 2021.

[2] Li W Z, Wang C D, Xiong H, et al. Graphsha: Synthesizing harder samples for class-imbalanced node classification[C]//Proceedings of the 29th ACM SIGKDD Conference on Knowledge Discovery and Data Mining. 2023: 1328-1340.

---

> ### Author Response · Authors · 2024-11-25
> **Response to the review comment**
>
> We thank you for your valuable time reviewing our manuscript, and for your appreciation, suggestions, questions and comments.
>
> > W1. The calculation of  value given in Theorem 1 is not rigorous, since the authors explore it in a certain range using predefined steps, which might not be the optimal value. The proof of Theorem 2 is also not convincing.
>
> Response: Thank you for your comment. The value of differentiation of LI with respect to $e_{bc}$ and $e_{wc}$ is a converging sequence at infinite values of $e_{bc}$ and $e_{wc}$ respectively. Thus even if we take any other range, we would get the same result approximately to some decimal places.
>
> For the proof of Theorem 2, we have added the appropriate references that make the subsequent steps clear.
>
> > W2. The chosen baselines are too old, and some key baselines are not discussed and compared, e.g., GraphENS, and GraphSHA, etc.
>
> Response: Thank you for your comment. Due to the time limit for the rebuttal period, we have now added comparison with GraphSHA[5] only as it is the most recent work on this topic as suggested by you, It can be clearly seen that LIMO is outperforming GraphSHA[5]. Please see the updated Table 2, 4, 6, and 7 for comparison in $cora$ and $citeseer$ dataset. We should be able to complete the rest of the datasets by camera-ready.
>
> The comparison between GraphSHA and LIMO on CoraLT is given here as reference. For more detailed experiment results, please refer to the updated version of the paper.
>
> | Imbalance Ratio | Setting  | LI         | Acc (%)   | AUC-ROC    | F1-Score   |
> |-----------------|----------|------------|-----------|------------|------------|
> | 0.6             | GraphSHA | 0.681      | 87.1      | 0.982      | 0.857      |
> |                 | **LIMO** | **0.97**   | **90.57** | **0.987**  | **0.987**  |
> | 0.5             | GraphSHA | 0.682      | 87.2      | 0.982      | 0.857      |
> |                 | **LIMO** | **0.97**   | **90.9**  | **0.987**  | **0.91**   |
> | 0.4             | GraphSHA | 0.682      | 87.07     | 0.982      | 0.857      |
> |                 | **LIMO** | **0.97**   | **90.23** | **0.987**  | **0.907**  |
> | 0.2             | GraphSHA | 0.679      | 87.37     | 0.9814     | 0.8596     |
> |                 | **LIMO** | **0.965**  | **90.60** | **0.9871** | **0.9096** |
> | 0.1             | GraphSHA | 0.6631     | 85.80     | 0.9733     | 0.8438     |
> |                 | **LIMO** | **0.9485** | **89.37** | **0.9858** | **0.8952** |
>
> >W3. No time and space complexity analyses are given to show the efficiency of the proposed model.
>
> Response: Thank you for pointing this out. Now we have added the time and space complexity analyses in section A.3 in the appendix.
>
> > Q1. The calculation of  value given in Theorem 1 is not rigorous, since the authors explore it in a certain range using predefined steps, which might not be the optimal value. The proof of Theorem 2 is also not convincing.
>
> Response:  Same as weakness W1
>
> >Q2. The chosen baselines are too old, and some key baselines are not discussed and compared, e.g., GraphENS [1], and GraphSHA [2], etc.
>
> Response:  Same as weakness W2
>
> > Q3. No time and space complexity analyses are given to show the efficiency of the proposed model.
>
> Response:  Same as weakness W3
>
> >Q4. The authors claim that the proposed model does not significantly increase the data volume. However, the authors synthesize nodes and edges in the graph, which indeed increases the data volume.
>
> Response: Thank you for pointing this out. We have removed the claim from the conclusion. We agree with your observation that LIMO increases the data volume due to the addition of synthetic nodes and edges, as is common with any SMOTE-based algorithm. This increase in data volume is a natural consequence of balancing the class distribution to improve model performance (and to some parts of increasing LI of the graph).
>
> > Q5. The caption in Figure 1 does not provide any information, and the authors did not cite references appropriately. For instance, SMOTE was not cited upon its first mention.
>
> Response: Thank you for pointing this out. This has been corrected now.
>
>
> > Q6. What labels are used in the Amazon datasets? The original datasets do not include labels for the user nodes.
>
> Response: Thank you for the question. In the Amazon datasets the labels are fraud and not fraud as used in this paper [3].
>
> Please refer to our revised manuscript to see the changes. We also provide a link to an anonymous repository with the updated code to include these changes.

---

> > ### Author Response · Authors · 2024-11-25
> > **Response to the review comment (Part 2)**
> >
> > References:
> >
> > [1] Park J, Song J, Yang E. Graphens: Neighbor-aware ego network synthesis for class-imbalanced node classification[C]//International conference on learning representations. 2021.
> >
> > [2] Li W Z, Wang C D, Xiong H, et al. Graphsha: Synthesizing harder samples for class-imbalanced node classification[C]//Proceedings of the 29th ACM SIGKDD Conference on Knowledge Discovery and Data Mining. 2023: 1328-1340.
> >
> > [3] Dou, Y., Liu, Z., Sun, L., Deng, Y., Peng, H., & Yu, P. S. (2020). Enhancing Graph Neural Network-based Fraud Detectors against Camouflaged Fraudsters. Proceedings of the 29th ACM International Conference on Information and Knowledge Management (CIKM '20), 315–324.

---

### Official Review · Reviewer_3wUc · 2024-10-29

**Soundness:** 3
**Presentation:** 3
**Contribution:** 2
**Rating:** 5
**Confidence:** 4

**Summary:**

This paper presents LIMO, an algorithm designed to tackle class imbalance in graph data. It works by adding synthetic edges between minority class nodes to increase Label Informativeness. It aims to improve the performance of GNNs on imbalanced datasets without significantly increasing the volume of data. The authors provide theoretical analysis and experimental results, showing LIMO outperforms existing methods, especially on heterophilous graphs.

**Strengths:**

* The paper is well-written and easy to understand.
* The paper focuses on the problem of class imbalance, a critical issue in many real-world applications.
* Rich experiments have been conducted on a variety of datasets, examining different aspects.

**Weaknesses:**

* The paper lacks a discussion of several recent and relevant baselines for addressing class imbalance in graphs. Notably, recent methods such as GraphENS [1], TAM [2], and GraphSHA [3] are not included in the experimental comparisons.
* The theoretical proof with t=1.31167628 is derived from simulation, but the paper does not provide a detailed explanation or justification of the simulation settings used. This lack of transparency limits the reliability of this theoretical result.
* While the paper theoretically claims that LI is directly proportional to GNN accuracy, experimental results do not consistently support this. For instance, in Figure 9, the correlation between LI and accuracy weakens when added_fraction > 0.5, especially at fraction = 0.75, where no strong positive correlation is observed.

[1] Park, J., Song, J., & Yang, E. (2021). Graphens: Neighbor-aware ego network synthesis for class-imbalanced node classification. In International conference on learning representations.
[2] Song, J., Park, J., & Yang, E. (2022, June). TAM: topology-aware margin loss for class-imbalanced node classification. In International Conference on Machine Learning (pp. 20369-20383). PMLR.
[3] Li, W. Z., Wang, C. D., Xiong, H., & Lai, J. H. (2023, August). Graphsha: Synthesizing harder samples for class-imbalanced node classification. In Proceedings of the 29th ACM SIGKDD Conference on Knowledge Discovery and Data Mining (pp. 1328-1340).

**Questions:**

* In the experiments, what does the imbalance ratio represent specifically set within the range of 0.1 to 0.6?
* Different dataset seems to show different performance. For example, in Table 10 when imbalance ratio = 0.1/0.2, while the LI of LIMO is the largest, the performances of RW and GS are better. Why?
* The experiment results can be further explained. For instance, in Figures 5 and 10, no significant correlation is observed once the fraction exceeds 0.25. Additionally, in Table 8, when the imbalance ratio is set to 0.1, ACC and F1-Score remain the same across four baseline methods.
* It is unclear whether the effectiveness stems more from well-synthesized node features or from the generated edges. Which one is more important? What are the appropriate $\delta$ values for interpolating node features?

**Details Of Ethics Concerns:**

* In the experiments, what does the imbalance ratio represent specifically set within the range of 0.1 to 0.6?
* Different dataset seems to show different performance. For example, in Table 10 when imbalance ratio = 0.1/0.2, while the LI of LIMO is the largest, the performances of RW and GS are better. Why?
* The experiment results can be further explained. For instance, in Figures 5 and 10, no significant correlation is observed once the fraction exceeds 0.25. Additionally, in Table 8, when the imbalance ratio is set to 0.1, ACC and F1-Score remain the same across four baseline methods.
* It is unclear whether the effectiveness stems more from well-synthesized node features or from the generated edges. Which one is more important? What are the appropriate $\delta$ values for interpolating node features?

---

> ### Author Response · Authors · 2024-11-25
> **Response to the review comment**
>
> We thank the reviewer for their valuable time reviewing our manuscript, and for your appreciation, suggestions, questions and comments.
>
> > W1. The paper lacks a discussion of several recent and relevant baselines for addressing class imbalance in graphs. Notably, recent methods such as GraphENS [1], TAM [2], and GraphSHA [3] are not included in the experimental comparisons.
>
> Response: Thank you for your comment. We have now added these references in the related works.
>
> Due to the time limit for the rebuttal period, we have now added comparison with GraphSHA[3] only as it is the most recent work on this topic as suggested by you, It can be clearly seen that LIMO is outperforming GraphSHA[3]. Please see the updated Table 2, 4, 6, and 7 for comparison in $cora$ and $citeseer$ dataset. We should be able to complete the rest of the datasets by camera-ready.
>
> The comparison between GraphSHA and LIMO on CoraLT is given here as reference. For more detailed experiment results, please refer to the updated version of the paper.
>
> | Imbalance Ratio | Setting  | LI         | Acc (%)   | AUC-ROC    | F1-Score   |
> |-----------------|----------|------------|-----------|------------|------------|
> | 0.6             | GraphSHA | 0.681      | 87.1      | 0.982      | 0.857      |
> |                 | **LIMO** | **0.97**   | **90.57** | **0.987**  | **0.987**  |
> | 0.5             | GraphSHA | 0.682      | 87.2      | 0.982      | 0.857      |
> |                 | **LIMO** | **0.97**   | **90.9**  | **0.987**  | **0.91**   |
> | 0.4             | GraphSHA | 0.682      | 87.07     | 0.982      | 0.857      |
> |                 | **LIMO** | **0.97**   | **90.23** | **0.987**  | **0.907**  |
> | 0.2             | GraphSHA | 0.679      | 87.37     | 0.9814     | 0.8596     |
> |                 | **LIMO** | **0.965**  | **90.60** | **0.9871** | **0.9096** |
> | 0.1             | GraphSHA | 0.6631     | 85.80     | 0.9733     | 0.8438     |
> |                 | **LIMO** | **0.9485** | **89.37** | **0.9858** | **0.8952** |
>
> > W2. The theoretical proof with t=1.31167628 is derived from simulation, but the paper does not provide a detailed explanation or justification of the simulation settings used. This lack of transparency limits the reliability of this theoretical result.
>
> Response: Thank you for the comment. We derive the value of $t$ numerically. We have provided the details of the derivation in appendix A.1. Additionally, we have now added a python notebook that simulates the derivation for $t$ in the anonymous repository in the abstract.
>
> > W3. While the paper theoretically claims that LI is directly proportional to GNN accuracy, experimental results do not consistently support this. For instance, in Figure 9, the correlation between LI and accuracy weakens when added_fraction > 0.5, especially at fraction = 0.75, where no strong positive correlation is observed.
>
> Response:
> Thank you for your comment and observation here. When the added fraction reaches higher values (e.g., fraction = 0.75), the additional edges tend to overfit the model to the training dataset. This overfitting reduces the model’s ability to generalize, weakening the correlation between LI and accuracy. Thus, the experimental results highlight a practical limitation where edge addition has diminishing returns and sometimes even negatively affects accuracy and LI. Adding more stability in this scenario is a focus of our future work.
>
> Questions:
> > Q1. In the experiments, what does the imbalance ratio represent specifically set within the range of 0.1 to 0.6?
>
> Response: Thank you for the question. Imbalance ratio is $\frac{\text{Number of minority nodes}}{\text{Number of majority nodes}}$, we have followed the imbalance ratios proposed in the GraphSMOTE [4]  paper.
>
> > Q2. Different dataset seems to show different performance. For example, in Table 10 (now Table 14)  when imbalance ratio = 0.1/0.2, while the LI of LIMO is the largest, the performances of RW and GS are better. Why?
>
> Response: Thank you for your question. When the imbalance ratio is set to 0.1 or 0.2, the addition of synthetic nodes is relatively small compared to the total number of nodes in other classes. This is because the setup allows adding only up to twice the number of minority nodes, which limits the impact of node addition on overall performance.
>
> Additionally, when edges are added (in case of Amazon (ALL), which is already a very dense dataset), they can sometimes cause the model to overfit the training dataset. This overfitting reduces the model's generalization capability, leading to a drop in performance when tested on unseen data. This explains why the performance of LIMO in this scenario does not surpass that of RW and GS, despite having the largest LI.

---

> > ### Author Response · Authors · 2024-11-25
> > **Response to the review comment (Part 2)**
> >
> > > Q3. The experiment results can be further explained. For instance, in Figures 5 and 10, no significant correlation is observed once the fraction exceeds 0.25. Additionally, in Table 8, when the imbalance ratio is set to 0.1, ACC and F1-Score remain the same across four baseline methods.
> >
> > Response: Thank you for the question.
> >
> > Figure 5 and 10: Thank you for pointing this out. In Figure 5, we observe that beyond a certain point, the decrease in LI no longer impacts accuracy. This is likely because, at very low LI values, the model performance is already saturated, and further reduction in LI does not significantly alter the underlying graph structure or its learning capability.
> >
> > Thank you for your observation regarding Table 8 (now Table 12). The identical ACC and F1-Score values across the four baseline methods when the imbalance ratio is set to 0.1 were obtained after multiple rounds and taken average. It is possible that, for this specific scenario, a very similar configuration was achieved across the methods, leading to comparable outcomes. We believe this reflects the inherent characteristics of the dataset and the model behaviors under the given imbalance ratio.
> >
> > > Q4. It is unclear whether the effectiveness stems more from well-synthesized node features or from the generated edges. Which one is more important? What are the appropriate  values for interpolating node features?
> >
> > Response: Thank you for the question. LIMO leverages SMOTE to generate “well-synthesized node features,” ensuring balanced representation for minority nodes. However, the addition of edges, which is specifically designed to enhance the LI, contributes significantly to the observed performance improvements.
> >
> > While SMOTE effectively interpolates node features, the additional performance gains beyond SMOTE are largely attributed to the generated edges. These edges play a crucial role in modifying the graph structure to optimize the Li property, which directly impacts the model’s ability to generalize and perform better on graph-based tasks.
> >
> > Please refer to our revised manuscript to see the changes. We also provide a link to an anonymous repository with the updated code to include these changes.
> >
> > References:
> >
> > [1] Park, J., Song, J., & Yang, E. (2021). Graphens: Neighbor-aware ego network synthesis for class-imbalanced node classification. In International conference on learning representations.
> >
> > [2] Song, J., Park, J., & Yang, E. (2022, June). TAM: topology-aware margin loss for class-imbalanced node classification. In International Conference on Machine Learning (pp. 20369-20383). PMLR.
> >
> > [3] Li, W. Z., Wang, C. D., Xiong, H., & Lai, J. H. (2023, August). Graphsha: Synthesizing harder samples for class-imbalanced node classification. In Proceedings of the 29th ACM SIGKDD Conference on Knowledge Discovery and Data Mining (pp. 1328-1340).
> >
> > [4] Zhao, Tianxiang, Xiang Zhang, and Suhang Wang. "Graphsmote: Imbalanced node classification on graphs with graph neural networks." Proceedings of the 14th ACM international conference on web search and data mining. 2021.
> >
> >
> > > Details of the ethics concerns:
> >
> > Response: Thank you for your questions. We answer all of these questions in the ‘questions’ section above.

---

### Official Review · Reviewer_z2eQ · 2024-11-01

**Soundness:** 2
**Presentation:** 2
**Contribution:** 1
**Rating:** 3
**Confidence:** 4

**Summary:**

To alleviate the class-imbalanced issue, this paper propose a novel algorithm,  Label Informativeness-based Minority Oversampling (LIMO), aiming to strategically synthesize minority nodes by augmenting edges to maximize label informativeness. And the experiments conducted on various homophilous and heterophilous benchmark datasets show the improvements compared with the baselines.

**Strengths:**

1. The paper is easy-to-follow.

**Weaknesses:**

1. The novelty is limited. It seems just simply inject SMOTE with label informativeness, however, the label informativeness is borrowed from previous work without clear explanation. Specifically, what is the meaning of the definition of LI in Eq.2? What situation will the LI(G) increase? And when will LI(G) decrease? More importantly, how does LI(G) influence the performance of class-imbalanced learning?

2. This paper fails to demonstrate its strengths compared with previous works. In lines 81-87, the authors claim that the proposed method take label informativeness to handle class-imbalanced issue. But  they fail to illustrate the limitations of previous works, and how the proposed method can tackle the limitations? Why does the previous works, like GraphSMOTE [1], GraphENS[2] and TAM[3], can not improve the label informativeness? What is the strengths of LIMO compared with these methods?

3. In line 152, during the node generation, LIMO need to find "its nearest neighbor $u$ in the same class". Does node $u$ belong to the labeled training set or the unlabeled nodes? If it comes from the training set, will it lead to overfitting when the number of minority classes is too small? If $u$ is an unlabeled node, how to get a reliable pseudo-label in the class-imbalanced scenario?

4. Lack of complexity analysis of the proposed method. In lines 11-13 of Algorithm 1, for each minority node $v$, LIMO need to determine the edge between $s$ and $w$, and $w \in \mathcal{V}-\{v\}$. The computation cost seems pretty high.

5. In line 216, "Specifically, it suggests that enhancing the connectivity between classes (increasing inter-class edges) can improve the informativeness of the labels". This conclusion seems to contradict graph homophily  [4] (the foundation of GNN), i.e., the connected nodes often come from the same class. In my opinion, we always want to add edges between two nodes from the same class, so that the nodes can aggregate more information from the same class, thus helping the node classification, just like GraphSMOTE[1] does. I have some doubt about the conclusion of line 216.

6. In addition, the baselines is out-of-date, there are many related works should be taken into comparison, for example, the oversampling methods like GraphENS[2] and GraphSHA[5], the pseudo-labeling method like GraphSR[6] and the loss adjustment method TAM[3].  The experimental setup is inappropriate.  In line 729, "When the minority class had fewer than three nodes, we allocated one node each for training, validation, and testing.",  It is unreasonable to have only one node for validation and testing. Why is the experimental setting not consistent with the previous method?



[1] Zhao T, Zhang X, Wang S. Graphsmote: Imbalanced node classification on graphs with graph neural networks[C]//Proceedings of the 14th ACM international conference on web search and data mining. 2021: 833-841.

[2] Park J, Song J, Yang E. Graphens: Neighbor-aware ego network synthesis for class-imbalanced node classification[C]//International conference on learning representations. 2021.

[3] Song J, Park J, Yang E. TAM: topology-aware margin loss for class-imbalanced node classification[C]//International Conference on Machine Learning. PMLR, 2022: 20369-20383.

[4] McPherson M, Smith-Lovin L, Cook J M. Birds of a feather: Homophily in social networks[J]. Annual review of sociology, 2001, 27(1): 415-44

[5] Li W Z, Wang C D, Xiong H, et al. Graphsha: Synthesizing harder samples for class-imbalanced node classification[C]//Proceedings of the 29th ACM SIGKDD Conference on Knowledge Discovery and Data Mining. 2023: 1328-1340.

[6] Zhou M, Gong Z. GraphSR: a data augmentation algorithm for imbalanced node classification[C]//Proceedings of the AAAI Conference on Artificial Intelligence. 2023, 37(4): 4954-4962.

**Questions:**

See weakness.

---

> ### Author Response · Authors · 2024-11-25
> **Response to the review comments**
>
> We thank the reviewer for their valuable time reviewing our manuscript, and for your appreciation, suggestions, questions and comments.
>
> > W1. The novelty is limited. It seems just simply inject SMOTE with label informativeness, however, the label informativeness is borrowed from previous work without clear explanation. Specifically, what is the meaning of the definition of LI in Eq.2? What situation will the LI(G) increase? And when will LI(G) decrease? More importantly, how does LI(G) influence the performance of class-imbalanced learning?
>
> Response: Thank you for your questions.
>
> Our contributions are as follows: To the best of our knowledge, existing SMOTE algorithms require learning over the input graph to introduce synthetic nodes to address the imbalance problem. In this work, we propose LIMO, a novel approach to other SMOTE techniques by generating synthetic edges specifically targeting LI. We formally establish the correlation between LI and model performance and validate our claims through extensive experiments. Our results demonstrate improved performance across all baselines, including the latest method, GraphSHA, as per your suggestion. We have updated the introduction section as well to motivate the use of LI.
> Additionally, we have updated Section 2.2 to clarify the relationship between changes in LI(G) and the conditions that lead to these changes.
>
> > W2. This paper fails to demonstrate its strengths compared with previous works. In lines 81-87, the authors claim that the proposed method take label informativeness to handle class-imbalanced issue. But they fail to illustrate the limitations of previous works, and how the proposed method can tackle the limitations? Why does the previous works, like GraphSMOTE [1], GraphENS[2] and TAM[3], can not improve the label informativeness? What are the strengths of LIMO compared with these methods?
>
> Response: Thank you for your insightful comment. Our approach fundamentally differs from merely building incrementally on previous methods. LIMO directly targets Label Informativeness (LI), which we have shown to have a strong correlation with model performance.
> The popular methods in the literature such as  GraphSMOTE [1], GraphENS[2], TAM[3] and GraphSHA[5]  require the model to be trained in order to mitigate the class imbalance for minority classes. This may also introduce bias in the nodes generated by the algorithms[1-4] towards the GNN backbone used such as GraphSAGE, GCN and GAT.
>
> While previous works may indirectly contribute to increasing LI despite not explicitly targeting it, LIMO directly targets increases LI directly and consistently achieves improved performance across all evaluated benchmarks, demonstrating its effectiveness in addressing the limitations of prior approaches. Please see the updated Table 2, 4, 6 and 7 for comparison in $cora$ and $citeseer$ dataset. We should be able to complete the rest of the datasets by camera-ready.
>
> > W3. In line 152, during the node generation, LIMO need to find "its nearest neighbor  in the same class". Does node  belong to the labeled training set or the unlabeled nodes? If it comes from the training set, will it lead to overfitting when the number of minority classes is too small? If  is an unlabeled node, how to get a reliable pseudo-label in the class-imbalanced scenario?
>
> Response: Thank you for the comment.  Node belongs to a labeled training set.
>
> In case the number of minority classes is too small, it might lead to overfitting as you have correctly pointed out. However, we should be able to mitigate that using mixup-like augmentation without label mixing from GraphENS[2]. Addressing this remains a focus of our future work.
>
>
> > W4. Lack of complexity analysis of the proposed method. In lines 11-13 of Algorithm 1, for each minority node , LIMO need to determine the edge between  and , and . The computation cost seems pretty high.
>
> Response:  Thank you for your comment. We have added the time and space complexity analyses in section A.3 in the appendix. Also, as noted in our conclusion, we acknowledge that LIMO is somewhat computationally expensive because of evaluation of new edges, and addressing this remains a focus of our future work.

---

> > ### Author Response · Authors · 2024-11-25
> > **Response to the review comments (Part 2)**
> >
> > > W5. In line 216, "Specifically, it suggests that enhancing the connectivity between classes (increasing inter-class edges) can improve the informativeness of the labels". This conclusion seems to contradict graph homophily [4] (the foundation of GNN), i.e., the connected nodes often come from the same class. In my opinion, we always want to add edges between two nodes from the same class, so that the nodes can aggregate more information from the same class, thus helping the node classification, just like GraphSMOTE[1] does. I have some doubt about the conclusion of line 216.
> >
> > Response: Thank you for your comment. As you correctly pointed out, graph homophily is commonly observed and often considered essential for the effectiveness of message-passing networks (GNNs). However, homophily is not strictly necessary for achieving good GNN performance. Certain forms of "good" heterophily, as discussed in [7], allow GNNs to perform effectively even on heterophilous graph data. In our paper, we leverage this insight by introducing inter-class edges, which enhance the label informativeness (LI) and, consequently, the models' performance, as demonstrated in our experiments.
> >
> > >W6. In addition, the baselines is out-of-date, there are many related works should be taken into comparison, for example, the oversampling methods like GraphENS[2] and GraphSHA[5], the pseudo-labeling method like GraphSR[6] and the loss adjustment method TAM[3]. The experimental setup is inappropriate. In line 729, "When the minority class had fewer than three nodes, we allocated one node each for training, validation, and testing.", It is unreasonable to have only one node for validation and testing. Why is the experimental setting not consistent with the previous method?
> >
> > Response: Thank you for your comment. Due to the time limit for the rebuttal period, we have now added comparison with GraphSHA[5] only as it is the most recent work on this topic as suggested by you. It can be clearly seen that LIMO outperforms GraphSHA[5]. Please see the updated Table 2, 4, 6 and 7 for comparison in $cora$ and $citeseer$ dataset. We should be able to complete the rest of the datasets by camera-ready.
> > The comparison between GraphSHA and LIMO on CoraLT is given here as reference. For more detailed experiment results, please refer to the updated version of the paper.
> >
> > | Imbalance Ratio | Setting  | LI         | Acc (%)   | AUC-ROC    | F1-Score   |
> > |-----------------|----------|------------|-----------|------------|------------|
> > | 0.6             | GraphSHA | 0.681      | 87.1      | 0.982      | 0.857      |
> > |                 | **LIMO** | **0.97**   | **90.57** | **0.987**  | **0.987**  |
> > | 0.5             | GraphSHA | 0.682      | 87.2      | 0.982      | 0.857      |
> > |                 | **LIMO** | **0.97**   | **90.9**  | **0.987**  | **0.91**   |
> > | 0.4             | GraphSHA | 0.682      | 87.07     | 0.982      | 0.857      |
> > |                 | **LIMO** | **0.97**   | **90.23** | **0.987**  | **0.907**  |
> > | 0.2             | GraphSHA | 0.679      | 87.37     | 0.9814     | 0.8596     |
> > |                 | **LIMO** | **0.965**  | **90.60** | **0.9871** | **0.9096** |
> > | 0.1             | GraphSHA | 0.6631     | 85.80     | 0.9733     | 0.8438     |
> > |                 | **LIMO** | **0.9485** | **89.37** | **0.9858** | **0.8952** |
> >
> > > W6 (cont.) "When the minority class had fewer than three nodes, we allocated one node each for training, validation, and testing.", It is unreasonable to have only one node for validation and testing. Why is the experimental setting not consistent with the previous method?
> >
> > Response This case was evident when we made our experiment setup consistent with GraphSMOTE[1] as shown in Table 2. However, now we have additionally used  the experiment setup similar to the setup of GraphENS[2] and GraphSHA[5] for datasets Cora LT (Table 2 and 4) and Citeseer LT (Table 6 and 7) which does not have this issue.
> >
> > Please refer to our revised manuscript to see the changes. We also provide a link to an anonymous repository with the updated code to include these changes.

---

> > > ### Author Response · Authors · 2024-11-25
> > > **Response to the review comments (Part 3)**
> > >
> > > References:
> > >
> > > [1] Zhao T, Zhang X, Wang S. Graphsmote: Imbalanced node classification on graphs with graph neural networks[C]//Proceedings of the 14th ACM international conference on web search and data mining. 2021: 833-841.
> > >
> > > [2] Park J, Song J, Yang E. Graphens: Neighbor-aware ego network synthesis for class-imbalanced node classification[C]//International conference on learning representations. 2021.
> > >
> > > [3] Song J, Park J, Yang E. TAM: topology-aware margin loss for class-imbalanced node classification[C]//International Conference on Machine Learning. PMLR, 2022: 20369-20383.
> > >
> > > [4] McPherson M, Smith-Lovin L, Cook J M. Birds of a feather: Homophily in social networks[J]. Annual review of sociology, 2001, 27(1): 415-44
> > >
> > > [5] Li W Z, Wang C D, Xiong H, et al. Graphsha: Synthesizing harder samples for class-imbalanced node classification[C]//Proceedings of the 29th ACM SIGKDD Conference on Knowledge Discovery and Data Mining. 2023: 1328-1340.
> > >
> > > [6] Zhou M, Gong Z. GraphSR: a data augmentation algorithm for imbalanced node classification[C]//Proceedings of the AAAI Conference on Artificial Intelligence. 2023, 37(4): 4954-4962.
> > >
> > > [7] Yao Ma, Xiaorui Liu, Neil Shah, and Jiliang Tang. Is homophily a necessity for graph neural networks?, 2023. URL https://arxiv.org/abs/2106.06134

---

### Official Review · Reviewer_11yk · 2024-11-01

**Soundness:** 3
**Presentation:** 3
**Contribution:** 3
**Rating:** 6
**Confidence:** 4

**Summary:**

The authors propose a new method for resolving class imbalances in graph-structured data based on a labeled informativeness-based minority oversampling method, called LIMO. By increasing the edges in a way that maximizes the amount of label information, LIMO strategically samples nodes of minority classes. And satisfactory results are obtained in node categorization dataset.

**Strengths:**

1. The authors propose Label Informativeness-based Minority Oversampling and theoretically establish the relationship between label informativeness and model prediction accuracy in GNN.
2. The authors analyze the effect of variation in the number of inter- and intra-class edges on LI.
3. The manuscript is logical and well-structured. The methodology section is clearly presented.

**Weaknesses:**

1. At the end of the first section, the authors show LIMO only in the figure, but there is no corresponding textual description in the text.
2. The currently proposed Label Informativeness-based Minority Oversampling (LIMO) approach is indeed a novel idea to solve the graph imbalance problem, but the core differences with existing approaches can be further highlighted in the introduction section or related work section such as 10.1109/IJCNN60899.2024.10650494, 10.1109/ICASSP48485.2024.10448064.
3. Discussion on the generalizability of LIMO is lacking in the manuscript, the authors only validate the performance effect of LIMO in GraphSAGE and lack discussion on LIMO in other GNNs such as Graph Convolutional Networks (GCN), Graph Attention Networks (GAT/ GAN). The authors need to provide a theoretical discussion of how LIMO might be generalized to other GNN architectures, or validated experimentally.
4. The experimental results are not comprehensive enough, and oversampling methods that specifically address the problem of graph data imbalance are missing from the comparison methods. Therefore, it is suggested that the authors should add methods that target graph data correlation in the experimental section, e.g., 10.1109/IJCNN60899.2024.10650494, 10.1109/ICASSP48485.2024.10448064.
5. Although the authors mention the inherent cost of the LIMO approach in generating and evaluating new edges in the summary section, the authors do not mention whether the added computational cost is within an acceptable range. The authors should provide actual runtime comparisons between LIMO and baseline methods, or a theoretical complexity analysis.

**Questions:**

1. Compared to existing oversampling methods based on graph data, what research gaps are addressed by the method proposed in this paper?
2. The authors proved Theorem 1 in the manuscript. Still, I am more interested in whether the variation of t for different problems affects the results of the experiments and how the optimal value of t is determined.
3. Is there a generalization of sacrificing significant computational cost to improve computational accuracy?

---

> ### Author Response · Authors · 2024-11-25
> **Response to the review comments**
>
> We thank the reviewer for their valuable time reviewing our manuscript, and for your appreciation, suggestions, questions and comments.
>
> > W1. At the end of the first section, the authors show LIMO only in the figure, but there is no corresponding textual description in the text.
>
> Response : Thank you for your comment.  We have updated the caption describing the figure.
>
> > W2 The currently proposed Label Informativeness-based Minority Oversampling (LIMO) approach is indeed a novel idea to solve the graph imbalance problem, but the core differences with existing approaches can be further highlighted in the introduction section or related work section such as 10.1109/IJCNN60899.2024.10650494, 10.1109/ICASSP48485.2024.10448064.
>
> Response: Thank you for your comment. We have now added a new paragraph in the Related Works section highlighting the working of the methods mentioned in the suggested papers and their differences with our approach.
>
> > W3. Discussion on the generalizability of LIMO is lacking in the manuscript, the authors only validate the performance effect of LIMO in GraphSAGE and lack discussion on LIMO in other GNNs such as Graph Convolutional Networks (GCN), Graph Attention Networks (GAT/ GAN). The authors need to provide a theoretical discussion of how LIMO might be generalized to other GNN architectures, or validated experimentally.
>
> Response: Thank you for your comment. We have now added GCN results in the paper in the Appendix section C.1. The performance improvement of LIMO compared to baselines is higher with GCN as the GNN backbone compared to GraphSAGE. This re-validates our claim. We shall be adding the experiments with GAT by camera ready.
>
> LIMO is a SMOTE technique for Graphs which is model agnostic. We have now verified its performance across 2 architectures i.e. GCN and GraphSAGE experimentally as advised. Therefore, we expect similar performance for all the message passing GNN architectures.
>
> > W4. The experimental results are not comprehensive enough, and oversampling methods that specifically address the problem of graph data imbalance are missing from the comparison methods. Therefore, it is suggested that the authors should add methods that target graph data correlation in the experimental section, e.g., 10.1109/IJCNN60899.2024.10650494, 10.1109/ICASSP48485.2024.10448064.
>
> Response: Thank you for the comment. Earlier we compared basic techniques (Oversampling, Reweight, SMOTE and EmbedSMOTE) in addition to GraphSMOTE [1] which serves as a widely recognized benchmark for evaluation. Now, we have added the comparison with a recent work GraphSHA [2]  that was recommended by other reviewers as well. The experiments in GraphSHA show its superiority compared to the methods in the papers suggested here.  We hope that these experiments are now sufficiently comprehensive.
> The comparison between GraphSHA and LIMO on CoraLT is given here as reference. For more detailed experiment results, please refer to the updated version of the paper.
>
> | Imbalance Ratio | Setting  | LI         | Acc (%)   | AUC-ROC    | F1-Score   |
> |-----------------|----------|------------|-----------|------------|------------|
> | 0.6             | GraphSHA | 0.681      | 87.1      | 0.982      | 0.857      |
> |                 | **LIMO** | **0.97**   | **90.57** | **0.987**  | **0.987**  |
> | 0.5             | GraphSHA | 0.682      | 87.2      | 0.982      | 0.857      |
> |                 | **LIMO** | **0.97**   | **90.9**  | **0.987**  | **0.91**   |
> | 0.4             | GraphSHA | 0.682      | 87.07     | 0.982      | 0.857      |
> |                 | **LIMO** | **0.97**   | **90.23** | **0.987**  | **0.907**  |
> | 0.2             | GraphSHA | 0.679      | 87.37     | 0.9814     | 0.8596     |
> |                 | **LIMO** | **0.965**  | **90.60** | **0.9871** | **0.9096** |
> | 0.1             | GraphSHA | 0.6631     | 85.80     | 0.9733     | 0.8438     |
> |                 | **LIMO** | **0.9485** | **89.37** | **0.9858** | **0.8952** |
> Additionally we are running other experiments, and we hope to be able to add to the camera ready if the experiments are complete.
>
>
> > W5. Although the authors mention the inherent cost of the LIMO approach in generating and evaluating new edges in the summary section, the authors do not mention whether the added computational cost is within an acceptable range. The authors should provide actual runtime comparisons between LIMO and baseline methods, or a theoretical complexity analysis.
>
> Response: Thank you for your comment. We have added the time and space complexity analyses in section A.3 in the appendix. Also, as noted in our conclusion, we acknowledge that LIMO is somewhat computationally expensive because of evaluation of new edges, and addressing this remains a focus of our future work.

---

> > ### Author Response · Authors · 2024-11-25
> > **Response to the review comments (Part 2)**
> >
> > Questions:
> > > Q1. Compared to existing oversampling methods based on graph data, what research gaps are addressed by the method proposed in this paper?
> >
> > Response: Thank you for the question. The methods in the literature such as GraphSMOTE [1], GraphSHA [2], TAM [3] and GraphENS [4] require the model to be trained in order to mitigate the class imbalance for minority classes. This may also introduce a bias in the nodes generated by the algorithms[1-4] towards the GNN backbone used such as GraphSAGE, GCN and GAT.  Our method performs oversampling without training any model on the dataset. This also reduces the overhead of training the models on the imbalanced dataset and then making the predictions.
> >
> > > Q2. The authors proved Theorem 1 in the manuscript. Still, I am more interested in whether the variation of t for different problems affects the results of the experiments and how the optimal value of t is determined.
> >
> > Response: Thank you for the question. $t$ as part of LI is a graph property. We have explained the detailed process to calculate the optimal value of $t$ in Theorem 1 in Appendix A.1. Additionally, we have now added a python notebook that simulates the derivation for $t$ in the anonymous repository in the abstract.
> > For the node classification problem:
> > 1. If \( t' < t \), adding intra-class edges such that their count falls within \([t', t]\) times the total intra-class edges for a specific class will decrease the LI.
> > 2. If \( t' > t \), adding inter-class edges between the minority class and another class, with their count falling within \([t, t']\) times the total intra-class edges of the minority class, will also decrease the LI.
> > Decrease in LI will reduce the model performance.
> >
> > We have added this explanation in the paper in section 4.
> > We did not understand what the reviewer meant by ‘different problems’. We can provide additional details if this is clarified.
> >
> > > Q3. Is there a generalization of sacrificing significant computational cost to improve computational accuracy?
> >
> > Response: Thank you for your question. We did not understand this question clearly.
> > However, we have added the time and space complexity analysis in appendix A.3. It remains in our scope of future work to improve on it.  We hope that this clarifies the reviewer’s doubt. We are open to discuss this further.
> >
> > Please refer to our revised manuscript to see the changes. We also provide a link to an anonymous repository with the updated code to include these changes.
> >
> >
> > References:
> >
> > [1] Tianxiang Zhao, Xiang Zhang, and Suhang Wang. 2021. GraphSMOTE: Imbalanced Node Classification on Graphs with Graph Neural Networks. In Proceedings of the 14th ACM International Conference on Web Search and Data Mining (WSDM '21). Association for Computing Machinery, New York, NY, USA, 833–841. https://doi.org/10.1145/3437963.3441720
> >
> > [2]  Wen-Zhi Li, Chang-Dong Wang, Hui Xiong, and Jian-Huang Lai. 2023. GraphSHA: Synthesizing Harder Samples for Class-
> > Imbalanced Node Classification. In Proceedings of the 29th ACM SIGKDD Conference on Knowledge Discovery and Data Mining (KDD '23). Association for Computing Machinery, New York, NY, USA, 1328–1340. https://doi.org/10.1145/3580305.3599374
> >
> > [3] Song J, Park J, Yang E. TAM: topology-aware margin loss for class-imbalanced node classification[C]//International Conference on Machine Learning. PMLR, 2022: 20369-20383.
> >
> > [4] Park J, Song J, Yang E. Graphens: Neighbor-aware ego network synthesis for class-imbalanced node classification[C]//International conference on learning representations. 2021.

---

> > > ### Comment · Reviewer_11yk · 2024-12-03
> > >
> > > Thank you for your detailed response. However, there are still the following problems in the paper. Although the paper includes more descriptions of existing research, I notice that these additions seem to stop at outlining previous studies rather than delving into the novelty of the LIMO method proposed in this paper compared to existing technologies. To fully demonstrate the value and contribution of LIMO, it is crucial to clearly articulate its fundamental differences from existing methods, as well as how it addresses problems that current approaches have not solved. The current discussion does not sufficiently explain the unique contributions of LIMO, nor does it explain how it provides improvements by increasing label informativeness. Therefore, I will maintain the current rating.

---

> > > > ### Author Response · Authors · 2024-12-03
> > > >
> > > > Thank you for your comment. We would like to clarify that our approach takes a fundamentally different direction from simply building incrementally on existing methods. LIMO is designed to directly target Label Informativeness (LI), which we have demonstrated to have a strong correlation with model performance.
> > > >
> > > > In contrast, popular methods in the literature, such as GraphSMOTE, GraphENS, TAM, and GraphSHA, rely on training the model to mitigate class imbalance for minority classes. This process can inadvertently introduce bias in the nodes generated by these algorithms, depending on the GNN backbone used, such as GraphSAGE, GCN, or GAT.
> > > >
> > > > While previous methods may indirectly contribute to increasing LI, LIMO explicitly and consistently achieves this, resulting in improved performance across all evaluated benchmarks. This underscores its effectiveness in addressing the limitations of prior approaches. For a detailed comparison, please refer to the updated Tables 2, 4, 6, and 7, which highlight results on the Cora and CiteSeer datasets. We are working towards completing results for the remaining datasets by the camera-ready submission deadline.

---

### Official Review · Reviewer_9tWm · 2024-11-04

**Soundness:** 2
**Presentation:** 2
**Contribution:** 3
**Rating:** 5
**Confidence:** 4

**Summary:**

The paper presents an oversampling technique for imbalance classification in graph classification settings. The method is based on the label informativeness as the criteria to augment the minority class samples. The method started with the regular non-graph SMOTE technique for generating new samples. Then it uses LI criteria to determine which nodes are connected to the new samples, in such a way that maximizes the LI criteria. The authors then demonstrate the performance of the proposed model in the real experiments.

**Strengths:**

1. Interesting ideas on using label informativeness to help tacking imbalance dataset.
2. The is built on top of SMOTE, a well known upsampling technique, and augment it to graph setting.
3. The experiments demonstrate the claimed benefit of the proposed approach.

**Weaknesses:**

1. Presentation. In many places, terminologies/abbreviations/symbols/equations are used without explaining them. For examples:
    - “IR”. (Page 2)
    - “SMOTE”. (Page 3)
    - “Eq (6)”. (Page 4)
    - “Eq (7)”. (Page 4)
    - etc
2. Motivation. The motivation behind using LI for upsampling is not explained clearly. Why is using LI a good idea to do upsampling? Why is having high LIs desirable? etc.
3. Soundness. The data augmentation by creating new nodes with new features using SMOTE technique makes sense. However, the way the proposed method augments the edges by attaching the new nodes to potentially unrelated nodes (not a neighbor of the original node), does not necessarily make sense. I get that the goal is to improve the LI metric. However, edges in the original graph represent certain relationship patterns. Attaching the new nodes to unrelated nodes will create new relationship patterns that do not exist in the original graph. These relationship patterns in the graph are something that is not yet explored by the model.

**Questions:**

1. Please address the weaknesses mentioned above.
2. In the evaluation, is the accuracy of the prediction computed using the augmented graph (after upsampling), or is it just based on the original graph?

---

> ### Author Response · Authors · 2024-11-25
> **Response to review comments**
>
> We thank the reviewer for their valuable time reviewing our manuscript, and for your appreciation, suggestions, questions and comments.
>
> > W1: Presentation. In many places, terminologies/abbreviations/symbols/equations are used without explaining them. For examples:
> “IR”. (Page 2)
> “SMOTE”. (Page 3)
> “Eq (6)”. (Page 4)
> “Eq (7)”. (Page 4)
> Etc
>
> Response: Thank you for bringing this to our attention. We have now added the related explanations at these and other places.
>
> > W2: Motivation. The motivation behind using LI for upsampling is not explained clearly. Why is using LI a good idea to do upsampling? Why is having high LIs desirable? Etc.
>
> Response: Thank you for the question. Empirical evidence has demonstrated a positive correlation between Label Informativeness (LI) and model performance [1]. Building on this, our work formally establishes this relationship. In general, enhancing the LI of a graph leads to improved model performance. To achieve this, we introduce synthetically generated nodes and edges to boost the graph's label informativeness.
> This has been added to the introduction section now to better reflect our motivation for using LI for upsampling.
>
> >W3: Soundness. The data augmentation by creating new nodes with new features using SMOTE technique makes sense. However, the way the proposed method augments the edges by attaching the new nodes to potentially unrelated nodes (not a neighbor of the original node), does not necessarily make sense. I get that the goal is to improve the LI metric. However, edges in the original graph represent certain relationship patterns. Attaching the new nodes to unrelated nodes will create new relationship patterns that do not exist in the original graph. These relationship patterns in the graph are something that is not yet explored by the model.
>
> Response: Thank you for the question. As you correctly pointed out, the goal of LIMO is to mitigate class-imbalance by improving the label informativeness. By doing this, we also intend that the augmented dataset contributes to the model learning to correctly perform the task of node classification. There is a possibility that the introduction of the new unrelated edges may lead to change in the relationship patterns in the original graph. However, the experimental evidence of creating the dataset using LIMO and then using it for the task with the GNN model demonstrates superior performance in contrast with the baselines. “Good” heterophily also contributes to the better accuracy of the GNN models [4].
> As our focus in this work was not specifically to preserve the graph structure, we have not verified the changes in the other graph properties and relationship pattern changes in the augmented graph  experimentally or theoretically.
>
> >Q1: Please address the weaknesses mentioned above.
>
> Response: Thank you for your comment. We have made the changes suggested by the reviewer in the previous section.
>
> >Q2: In the evaluation, is the accuracy of the prediction computed using the augmented graph (after upsampling), or is it just based on the original graph?
>
> Response: Thank you for the question. It is calculated on the augmented graph similar to other works like GraphSMOTE [2],  GraphSHA[3].
>
> Please refer to our revised manuscript to see the changes. We also provide a link to an anonymous repository with the updated code to include these changes.
>
> References:
>
> [1] Oleg Platonov, Denis Kuznedelev, Artem Babenko, and Liudmila Prokhorenkova. Characterizing graph datasets for node classification: Homophily-heterophily dichotomy and beyond, 2024. URL https://arxiv.org/abs/2209.06177.
>
> [2] Zhao, Tianxiang, Xiang Zhang, and Suhang Wang. "Graphsmote: Imbalanced node classification on graphs with graph neural networks." Proceedings of the 14th ACM international conference on web search and data mining. 2021.
>
> [3] Wen-Zhi Li, Chang-Dong Wang, Hui Xiong, and Jian-Huang Lai. 2023. GraphSHA: Synthesizing Harder Samples for Class-Imbalanced Node Classification. In Proceedings of the 29th ACM SIGKDD Conference on Knowledge Discovery and Data Mining (KDD '23). Association for Computing Machinery, New York, NY, USA, 1328–1340. https://doi.org/10.1145/3580305.3599374
>
> [4] Yao Ma, Xiaorui Liu, Neil Shah, and Jiliang Tang. Is homophily a necessity for graph neural networks?, 2023. URL https://arxiv.org/abs/2106.06134

---

> > ### Comment · Reviewer_9tWm · 2024-12-01
> >
> > Thanks to the authors for the reply.
> >
> > > There is a possibility that the introduction of the new unrelated edges may lead to change in the relationship patterns in the original graph. However, the experimental evidence of creating the dataset using LIMO and then using it for the task with the GNN model demonstrates superior performance in contrast with the baselines. “Good” heterophily also contributes to the better accuracy of the GNN models [4]. As our focus in this work was not specifically to preserve the graph structure, we have not verified the changes in the other graph properties and relationship pattern changes in the augmented graph experimentally or theoretically.
> >
> > I understand that having desirable good properties in the graph may improve the accuracy of GNNs. However, introducing those may change the graph classification itself and may not generalize well into new cases. One way to test the generalization is to have a full set a new graph (could be based on partitioning the graph), and do prediction on this graph without augmentation, and see if the model also perform well in this new graph.

---

> > > ### Author Response · Authors · 2024-12-01
> > >
> > > Thank you for the comment.
> > >
> > > We shall definitely take a look at the changing graph properties and its effect on generalization in the future. Thank you for suggesting a way to test it as well. We hope to add these experimental results by the camera ready deadline, as we might not have a scope to finish these experiments before the rebuttal time ends.
> > >
> > > In the current scenario, we have tried to replicate the experiment setup of both the base paper GraphSMOTE and the state of the art GraphSHA paper.
> > > Additionally papers like GraphSMOTE and GraphSHA also change graph properties in an indirect way. This is evident from our experiments, as seen in the column titled 'LI' in tables 2, 4, 6, 7.

---

### Author Response · Authors · 2024-12-02

Dear Reviewers,

We thank you for taking time to review our paper. In response to your valuable feedback, we have revised the manuscript accordingly and conducted additional experiments to address the points raised. These changes have been carefully documented, and we believe your insights have significantly enhanced the quality of our work.

Should you have any further comments or suggestions, we would be most grateful to hear them. We appreciate your time and look forward to hearing from you.

Thank you.

Authors

---

### Author Response · Authors · 2024-12-04

Dear Reviewers and Area Chairs,

As the discussion period nears its conclusion, we would like to bring to your attention the significant revisions and improvements made to our manuscript in response to the reviewers’ feedback.

First, we would like to emphasize that all the points raised by the reviewers were addressable, and we have revised the manuscript accordingly. Specifically, we conducted additional experiments to benchmark our approach, LIMO, against state-of-the-art methods, including GraphSHA. These results, now included in the revised manuscript, demonstrate that LIMO consistently outperforms GraphSHA across various datasets and tasks.

While we understand that the initial version of our manuscript received low scores, these scores do not take into account the substantial updates made in the revised version. Furthermore, during the discussion period, we received comments from only one out of the five reviewers, leaving us uncertain whether the improvements have been fully evaluated.

ICLR provides authors with a unique opportunity to revise their submissions in response to reviewer feedback, setting it apart from many other conferences. We have taken this opportunity seriously and have addressed all the reviewers' concerns to the best of our ability. The revisions include new experimental results, clarifications to the methodology, and improvements in presentation, all of which are highlighted in blue in the revised manuscript for ease of review.

We kindly request that the Area Chair and reviewers consider the revised paper and our detailed responses before making a final decision. If the paper is ultimately rejected, we would deeply appreciate constructive feedback specific to the revised version. This will help us refine the work further and avoid resubmitting what has already been improved here.

Thank you for your time, feedback, and consideration.

Authors

---

### Meta-Review · Area_Chair_zFMM · 2024-12-19

**Metareview:**

The paper received five reviews with ratings of 5, 6, 3, 5, and 5. The reviewers raised several major concerns regarding the paper, including limited novelty, insufficient experimental evaluation with outdated baseline methods, unclear motivation and discussion regarding the contributions of LIMO, lack of complexity analysis, and insufficient clarity in the presentation. Despite the authors’ rebuttal, the reviewers remained unconvinced and retained their initial ratings. Consequently, this paper is recommended for rejection.

**Additional Comments On Reviewer Discussion:**

A few reviewers engaged in discussions with the authors but ultimately decided to maintain their original rating scores.

---

### Decision · Program_Chairs · 2025-01-22

Reject